# UNLOCKING STATE-TRACKING IN LINEAR RNNS THROUGH NEGATIVE EIGENVALUES

**Riccardo Grazzi**[*♡], **Julien Siems**[*◇], **Arber Zela**[◇],
**Jörg K.H. Franke**[◇], **Frank Hutter**[◇♣], **Massimiliano Pontil**[♡♠]
Equal contribution[*], CSML, Istituto Italiano di Tecnologia[♡], University of Freiburg[◇],
ELLIS Institute Tübingen[♣], AI Centre, University College London[♠]
riccardograzzi4@gmail.com      juliensiems@gmail.com

## ABSTRACT

Linear Recurrent Neural Networks (LRNNs) such as Mamba, RWKV, GLA, mL-STM, and DeltaNet have emerged as efficient alternatives to Transformers for long sequences. However, both Transformers and LRNNs struggle to perform state-tracking, which may impair performance in tasks such as code evaluation. In one forward pass, current architectures are unable to solve even parity, the simplest state-tracking task, which non-linear RNNs can handle effectively. Recently, Sarrof et al. (2024) demonstrated that the failure of LRNNs like Mamba to solve parity stems from restricting the value range of their diagonal state-transition matrices to $[0, 1]$ and that incorporating negative values can resolve this issue. We extend this result to non-diagonal LRNNs such as DeltaNet. We prove that finite precision LRNNs with state-transition matrices having only positive eigenvalues cannot solve parity, while non-triangular matrices are needed to count modulo 3. Notably, we also prove that LRNNs can learn any regular language when their state-transition matrices are products of identity minus vector outer product matrices, each with eigenvalues in the range $[-1, 1]$. Our experiments confirm that extending the eigenvalue range of Mamba and DeltaNet to include negative values not only enables them to solve parity but consistently improves their performance on state-tracking tasks. We also show that state-tracking enabled LRNNs can be pretrained stably and efficiently at scale (1.3B parameters), achieving competitive performance on language modeling and showing promise on code and math tasks.

## 1 INTRODUCTION

Transformer architectures (Vaswani et al., 2017) have revolutionized NLP but scale quadratically in sequence length, posing computational challenges for long sequences. To address this, Linear Recurrent Neural Networks (LRNNs) have emerged as promising alternatives that offer linear scaling while maintaining competitive performance (Gu & Dao, 2024; Dao & Gu, 2024; Yang et al., 2024a; Peng et al., 2023; Deletang et al., 2023; Sun et al., 2024; Beck et al., 2024). LRNNs update their state via matrix-vector products with structured and often input-dependent state-transition matrices. The structure of the state-transition matrices largely determines the expressivity of LRNNs. While successful models like Mamba (Gu & Dao, 2024) and GLA (Yang et al., 2024a) use diagonal matrices (diagonal LRNN) which only mix tokens along the sequence dimension, recent work explores more complex forms. Notably, non-diagonal matrices using generalized Householder (GH) transformations,

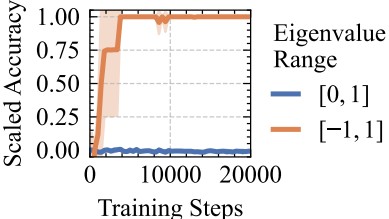

Figure 1: Extending the eigenvalue range of the state transition matrices of diagonal LRNNs improves performance from random guessing (range $[0, 1]$) to perfect score (range $[-1, 1]$) on learning parity. Trained on sequences up to length 40; Tested on lengths 40–256 (3 seeds).

defined as $I - uu^\top$ where $u$ is a learnable vector and $I$ is the identity, enable models like DeltaNet (Schlag et al., 2021; Yang et al., 2024b) and TTT-Linear (Sun et al., 2024) to achieve richer expressiveness through simultaneous token-channel mixing while maintaining efficiency.

Surprisingly, both Transformers and current LRNNs face a fundamental limitation: they struggle to learn how to track the state of even simple finite-state machines from sequences of state-transitions (Deletang et al., 2023). This limitation may impair performance on tasks such as entity tracking in narratives, handling nested structures in code, and reasoning tasks that can benefit from maintaining and updating an internal state over time (Merrill et al., 2024). Even the simplest state-tracking task, computing the parity of a sequence of bits, cannot be solved by modern architectures, while non-linear RNNs like LSTM (Hochreiter & Schmidhuber, 1997) and sLSTM (Beck et al., 2024) can effectively track the state of any finite state machine. However, parallelizing non-linear RNNs across the sequence length presents significant challenges (Lim et al., 2024; Gonzalez et al., 2024).

Recently, Sarrof et al. (2024) demonstrated that the inability of diagonal LRNNs to solve the *parity* problem stems from the fact that the eigenvalues of their state-transition matrices are constrained to be positive. Specifically, they proved that finite precision diagonal LRNNs with exclusively positive real eigenvalues, cannot solve the parity problem in one forward pass for sequences of arbitrary length. However, their work did not provide empirical evidence showing that diagonal LRNNs with negative eigenvalues can be successfully trained to overcome this limitation. We prove that the same limitation also affects LRNNs with non-diagonal state-transition matrices, and further prove that additionally, non-triangular matrices are necessary to solve the more challenging task of modular counting (when the modulus is not a power of two). Our findings also apply to the GH matrices used by DeltaNet, as they share the same eigenvalue limitations. To overcome this, we propose a simple yet powerful solution: extend the range of possible eigenvalues from $[0, 1]$ to $[-1, 1]$. This change enables state-tracking and significantly improves the expressivity of LRNNs without compromising their efficiency and training stability. As illustrated in Figure 1, it allows diagonal LRNNs to learn parity successfully. The code for part of our experiments is available at https://github.com/automl/unlocking_state_tracking

In summary, we make the following *contributions:*
1. We prove that any finite precision LRNN with only positive real eigenvalues in the state-transition matrices (most LRNNs used in practice) cannot solve parity at arbitrary sequence lengths (Theorem 1), while non-triangular matrices are also required to learn counting modulo 3 (Theorem 2).
2. By extending the eigenvalue range, we significantly improve the state-tracking capabilities of LRNNs. We prove that LRNNs with state-transition matrices formed by products of generalized Householder (GH) matrices, each with eigenvalues in the range $[-1, 1]$, can learn any regular language (Theorem 4), in some cases with just one layer (Theorem 3). Notably, this range extension allows LRNNs using just one GH matrix (like DeltaNet), to learn substantially harder tasks, as the repeated composition of permutations of two (over n) elements, compared to diagonal LRNNs.
3. We show that the eigenvalue range of Mamba and DeltaNet can be extended to $[-1, 1]$ without compromising efficiency or training stability. We test the modified methods on parity, modular arithmetic, and permutation composition, demonstrating improved state-tracking performance.
4. We pre-train modified versions of DeltaNet and Mamba (up to 1.3B parameters) and show that they reach performance comparable to the original models on generative language modeling tasks, while DeltaNet shows improved perplexity on code and math datasets.

## 2 RELATED WORK

**Linear RNNs.** Linear RNNs encompass state-space models and causal, linear attention mechanisms. State-space models, originally used for continuous dynamical systems, inspired LRNN variants like S4 (Gu et al., 2022) and H4 (Fu et al., 2021) (see Tiezzi et al. (2024) for a survey). Recent advancements, such as Mamba (Gu & Dao, 2024; Dao & Gu, 2024), introduced input-dependent gating of the hidden state, significantly improving language modeling performance. Concurrently, linear attention has emerged as an alternative to classical softmax attention, with Katharopoulos et al. (2020) demonstrating that causal linear attention Transformers can be reformulated as RNNs with linear scaling in sequence length. Building on this, Yang et al. (2024a) proposed Gated Linear Attention (GLA), adding a gating mechanism similar to Mamba, while DeltaNet (Schlag et al., 2021; Yang et al., 2024b) and TTT-Linear (Sun et al., 2024) explored more expressive recurrences with non-diagonal state-transition matrices. Beck et al. (2024) recently proposed xLSTM, a successor to LSTM (Hochreiter & Schmidhuber, 1997) which combines non-linear and linear RNNs.

**Expressivity Results.** Several studies have explored the expressive power of Transformers and RNNs (see e.g. (Merrill et al., 2020; Strobl et al., 2024; Bhattamishra et al., 2024)). Here, we

focus on the ones most relevant to our work. While Hahn (2020) proved that Transformers cannot model periodic languages such as parity, see also (Bhattamishra et al., 2020, Lemma C.4), and some context-free languages at arbitrary sequence lengths, Liu et al. (2023) demonstrated that Transformers can learn shortcut solutions for *solvable* finite state automata, though these solutions lack generalizability to arbitrary sequence lengths and perform poorly out-of-distribution. Unlike RNNs, the high parallelizability of Transformers prevents them from learning *unsolvable* finite state automata (Merrill & Sabharwal, 2023). These findings typically use techniques from algebraic formal language theory (we refer to Liu et al. (2023) for a short tutorial) and circuit complexity, using the *log-precision assumption* and a number of layers scaling linearly or logarithmically with sequence length. While earlier research established Transformers' Turing completeness, it relied on either arbitrary precision (Pérez et al., 2021) or arbitrary depth and weight sharing (Giannou et al., 2023). Diagonal LRNNs can simulate any RNN with infinite depth (Gu et al., 2021) and approximate regular enough functions when the state dimension grows linearly with sequence length (Orvieto et al., 2024). However, things change when depth and state size are fixed. Merrill et al. (2024) proved that finite-depth diagonal LRNNs, like Transformers, struggle to learn unsolvable finite state automata when restricted to log-precision arithmetic. The work by Fan et al. (2024) highlights a similar limitation, while in a finite precision setting, Sarrof et al. (2024) showed that diagonal LRNNs with positive values in the state-transition matrix, while capable of learning all star-free languages, cannot solve even the simple *parity* problem, a non-star-free language recognizable by an automaton with two states. However, their analysis was limited to the diagonal case and they did not test the benefit of negative eigenvalues in practice. Using a continuous time framework, also Cirone et al. (2025) pointed out the limitations of diagonal state transition matrices. Irie et al. (2021; 2023) empirically showed how state-tracking can be enabled by modifying DeltaNet as a fast weight programmer (Schmidhuber, 1992), but this makes its recurrence non-linear, hence hard to parallelize. Unlike previous work, we demonstrate that non-diagonal LRNNs like DeltaNet can achieve robust state-tracking through a minimal modification while maintaining efficient large-scale training.

## 3 BACKGROUND

### 3.1 LINEAR RECURRENT NEURAL NETWORKS (LRNNs)

We describe LRNNs using notation inspired by Sarrof et al. (2024), focusing on the core linear recurrences while abstracting away the non-linear computations for each token. LRNNs are stacks of layers that share a common structure but have distinct learnable parameters. Each layer takes input vectors $\boldsymbol{x}_1, \ldots, \boldsymbol{x}_t \in \mathbb{R}^l$ (outputs of the previous layer) and outputs $\hat{\boldsymbol{y}}_1, \ldots, \hat{\boldsymbol{y}}_t \in \mathbb{R}^p$ as:

$$\boldsymbol{H}_i = \boldsymbol{A}(\boldsymbol{x}_i)\boldsymbol{H}_{i-1} + \boldsymbol{B}(\boldsymbol{x}_i), \quad \hat{\boldsymbol{y}}_i = \mathrm{dec}(\boldsymbol{H}_i, \boldsymbol{x}_i), \quad \text{for all } i \in \{1, \ldots, t\},$$
$$\boldsymbol{H}_0 \in \mathbb{C}^{n \times d}, \quad \boldsymbol{A} : \mathbb{R}^l \to \mathbb{C}^{n \times n}, \quad \boldsymbol{B} : \mathbb{R}^l \to \mathbb{C}^{n \times d}, \quad \mathrm{dec} : \mathbb{C}^{n \times d} \times \mathbb{R}^l \to \mathbb{R}^p \tag{1}$$

Here, $\boldsymbol{A}, \boldsymbol{B}$ and dec are learnable, generally non-linear functions, with dec usually containing a feed-forward neural network. This definition encompasses most LRNN variants, which differ in the form of $\boldsymbol{A}, \boldsymbol{B}$ and dec. Table 1 illustrates how three popular LRNNs fit this framework. For other architectures see (Yang et al., 2024b, Table 4). Additional details on the notation are in Appendix A.1.

Table 1: Instances of LRNN layers in (1), where $\boldsymbol{\alpha}_t = \mathrm{sigmoid}(\boldsymbol{W}_\alpha \boldsymbol{x}_t)$, $\boldsymbol{\Delta}_t = \mathrm{softplus}(\boldsymbol{W}_\Delta \boldsymbol{x}_t)$, $\beta_t = \mathrm{sigmoid}(\boldsymbol{w}_\beta^\top \boldsymbol{x}_t)$, while $\boldsymbol{q}_t, \boldsymbol{k}_t \in \mathbb{R}^n, \boldsymbol{v}_t \in \mathbb{R}^d$ are output of learnable functions of $\boldsymbol{x}_t$. Also, $\psi : \mathbb{R}^d \to \mathbb{R}^d$ is another learnable function usually containing an MLP and a normalization, while $\boldsymbol{W}_1 \in \mathbb{R}^{n \times d}$, $\boldsymbol{W}_\Delta \in \mathbb{R}^{d \times l}$, $\boldsymbol{W}_\alpha \in \mathbb{R}^{n \times l}$, $\boldsymbol{w}_\beta \in \mathbb{R}^l$ and $\boldsymbol{w}_2 \in \mathbb{R}^d$ are learnable parameters. For simplicity, we omitted 1D convolutions. For Mamba, the matrices in the first two columns represent the recurrence for the i-th row of $\boldsymbol{H}_t$ and we set $\boldsymbol{k}_t = (k_{t,1}, \ldots, k_{t,n})^\top$, $\boldsymbol{W}_1 = (\boldsymbol{w}_{1,1}, \ldots, \boldsymbol{w}_{1,n})^\top$, and $l = d$.

| | $\boldsymbol{A}(\boldsymbol{x}_t)$ | $\boldsymbol{B}(\boldsymbol{x}_t)$ | $\mathrm{dec}(\boldsymbol{H}_t, \boldsymbol{x}_t)$ |
|---|---|---|---|
| **Mamba** | $\mathrm{Diag}\left(\exp\left(-\boldsymbol{\Delta}_t \odot \exp(\boldsymbol{w}_{1,i})\right)\right)$ | $k_{t,i}\boldsymbol{\Delta}_t \odot \boldsymbol{x}_t$ | $\psi(\boldsymbol{H}_t^\top \boldsymbol{q}_t + \boldsymbol{w}_2 \odot \boldsymbol{x}_t)$ |
| **GLA** | $\mathrm{Diag}\left(\boldsymbol{\alpha}_t\right)$ | $\boldsymbol{k}_t \boldsymbol{v}_t^\top$ | $\psi(\boldsymbol{H}_t^\top \boldsymbol{q}_t)$ |
| **DeltaNet** | $\boldsymbol{I} - \beta_t \boldsymbol{k}_t \boldsymbol{k}_t^\top$ | $\beta_t \boldsymbol{k}_t \boldsymbol{v}_t^\top$ | $\psi(\boldsymbol{H}_t^\top \boldsymbol{q}_t)$ |

The *state-transition matrices* $\boldsymbol{A}(\boldsymbol{x}_t)$ are typically diagonal or generalized Householder (GH), i.e., identity minus vector outer product, as shown in Table 1, to enable efficient matrix-vector products on modern hardware. These matrices consistently have eigenvalues (and norm) in the range $[0, 1]$.

## 3.2 FORMAL LANGUAGE THEORY

**Finite State Automata and Regular Languages.** A (deterministic) finite state automaton (FSA) is a tuple $\mathcal{A} = (\Sigma, Q, q_0, \delta)$ where $\Sigma$ is a finite set of letters called alphabet, $Q$ is a finite set of states, $q_0 \in Q$ is the starting state and $\delta : Q \times \Sigma \to Q$ is the state-transition function (see Hopcroft & Ullman, 2001, for an introduction). We define the set $\Sigma^*$, whose elements are sequences called words, as the smallest superset of $\Sigma$ that contains the empty word $\varepsilon$ and is closed under word concatenation. We extend the state-transition function to $\delta : Q \times \Sigma^* \to Q$ by defining $\delta(q, \varepsilon) = q$ and $\delta(q, \boldsymbol{w}) = \delta(\delta(q, w_1 \ldots w_{i-1}), w_i)$ for any $\boldsymbol{w} = w_1 \ldots w_i \in \Sigma^*$ with $i \geq 2$. We say that $\delta(q_0, \boldsymbol{w})$ is the state that $\mathcal{A}$ reaches after reading the word $\boldsymbol{w} \in \Sigma^*$. A *language* $L \subseteq \Sigma^*$ is said to be recognized by $\mathcal{A}$ if there exists a recognizing set $R \subseteq Q$ such that $L = \{\boldsymbol{w} \in \Sigma^* : \delta(q_0, \boldsymbol{w}) \in R\}$. Regular languages are the ones that can be recognized by an FSA. Given an FSA $\mathcal{A}$, the set $\mathcal{T}(\mathcal{A}) = \{\delta(\cdot, \boldsymbol{w}) : \boldsymbol{w} \in \Sigma^*\}$ of functions $\rho : Q \to Q$, together with the function composition operation forms a *monoid* called *transition monoid*, i.e. it is associative, closed and contains the identity $\delta(\cdot, \varepsilon)$. This monoid has a finite number of elements, since $|Q| < \infty$. Moreover, if $\delta(\cdot, w)$ is bijective for every $w \in \Sigma$, then $\mathcal{T}(\mathcal{A})$ forms a *group*, i.e. it contains the inverse of each element.

**State-Tracking and Monoid Word Problems.** State-tracking is the problem of determining the state of a system only by observing a sequence of updates applied to it. Formally, it can be expressed as a *monoid word problem* (Merrill et al., 2024), where given a monoid $(M, \cdot)$ ($M$ is the set and $\cdot$ is the associative operation), we want to send words $m_1 \ldots m_t \in M^*$, describing the sequence of updates, to their product $m_1 \cdot m_2 \cdots m_t \in M$, representing the state of the system after the updates. If $M$ is finite there is a corresponding FSA $(M, M, e, \delta)$ that solves the word problem, where the starting state is $e$ (the identity element), and the transition function is $\delta(m_1, m_2) = m_2 \cdot m_1$ for $m_1, m_2 \in M$. In this work, we focus on group word problems, i.e. problems where the monoid is also a group. In particular, on the cyclic group $\mathbb{Z}_m$, i.e. addition modulo $m$, and the symmetric group $S_m$, i.e. the group of permutations on $m$ elements. Parity is equivalent to the $S_2$ word problem, while many state-tracking problems such as tracking chess moves or code evaluation, can be shown to be harder than the $S_5$ word problem, which cannot be solved by Transformers and diagonal LRNNs even in log-precision for arbitrary word lengths (Merrill et al., 2024; Merrill & Sabharwal, 2023).

**One LRNN Layer is an automaton.** Given an alphabet $\Sigma \subset \mathbb{N}$, we can view one layer of an LRNN in (1) as the automaton $\mathcal{A}_{\text{lin}} = (\Sigma, \mathcal{H}, \boldsymbol{H}_0, \delta_{\text{lin}})$, where $\delta_{\text{lin}}(\boldsymbol{H}, w) = \boldsymbol{A}(w)\boldsymbol{H} + \boldsymbol{B}(w)$, which is extended as we saw previously[1], and $\mathcal{H} = \{\delta_{\text{lin}}(\boldsymbol{H}_0, \boldsymbol{w}) : \boldsymbol{w} \in \Sigma^*\} \subseteq \mathbb{R}^{n \times d}$. We say that an LRNN layer in (1) *implements* the FSA $\mathcal{A} = (\Sigma, Q, q_0, \delta)$ if $\mathcal{A}_{\text{lin}}$ can mimic the state transitions of $\mathcal{A}$[2]. Formally, if there exists a surjective function $g : \mathcal{H} \to Q$, such that for any $\boldsymbol{H} \in \mathcal{H}$, $w \in \Sigma$ $\delta(g(\boldsymbol{H}), w) = g(\delta_{\text{lin}}(\boldsymbol{H}, w)) = g(\boldsymbol{A}(w)\boldsymbol{H} + \boldsymbol{B}(w))$. Every language $L$ recognized by $\mathcal{A}$ can also be recognized by this LRNN layer with a sufficiently powerful dec. In particular if $R \subseteq Q$ is the recognizing set for $L$ and $q_0 = g(\boldsymbol{H}_0)$, then the decoder $\text{dec}(\boldsymbol{H}_t, w_t) = \mathbf{1}\{g(\boldsymbol{H}_t) \in R\}$, will correctly determine if $\boldsymbol{w} \in L$. Therefore, implementing $\mathcal{A}$ is at least as hard as recognizing $L$. A principal goal of this work is to show that current LRNNs cannot recognize simple languages such as parity (negative results) while appropriate modifications to the state-transition matrices, enable LRNNs to implement broader classes of FSA (positive results), with certain classes of FSA requiring a single layer. Note, that while LRNNs with one layer can recognize any regular language, the state transition matrices might not fit into the structure imposed by current LRNNs, such as those in Table 1 (see Appendix A.3 for more details).

# 4 THEORETICAL ANALYSIS

## 4.1 LIMITATIONS OF CURRENT LRNNS

In this section, we describe how positive eigenvalues and non-triangular state transition matrices limit LRNNs state-tracking capabilties. In particular, we focus on parity and modular addition. The parity $y_t \in \{0, 1\}$ of a sequence of ones and zeros $x_1 \ldots x_t \in \{0, 1\}^t$ is 1 if the total number of ones in the sequence is odd, and 0 if it's even. Equivalent to addition modulo 2, it can be computed by summing the values in the input sequence and then applying the modulo 2 function: $y_t = (\sum_{i=1}^{t} x_i) \bmod 2$. This solution can be implemented by an LRNN with one layer and scalar

---

[1] We let $\delta_{\text{lin}} : \mathbb{R}^{n \times d} \times \Sigma \to \mathbb{R}^{n \times d}$ and extend it to $\delta_{\text{lin}} : \mathbb{R}^{n \times d} \times \Sigma^* \to \mathbb{R}^{n \times d}$, then we define $\mathcal{H}$.

[2] This definition is equivalent to that of FSA homomorphism, see (Maler & Pnueli, 1994, Definition 3).

Figure 2: *Parity requires negative eigenvalues.* States of one-layer LRNNs with the sequence $1111\ldots$ as input. If the eigenvalues of $\mathbf{A}(1)$ are nonnegative, the states either diverge or converge monotonically, and so, for large enough $t$ and in finite precision, cannot be distinguished. In contrast, the LRNN with $a(1) = -1$ alternates between two states like the parity automaton.

states by setting $\mathbf{A}(x_t) = 1$, $\mathbf{B}(x_t) = x_t$, $\mathbf{H}_0 = 0$, and $\mathrm{dec}(\mathbf{H}_t, x_t) = \mathbf{H}_t \bmod 2$ in (1). However, implementing such a solution with finite precision presents an issue: the state $h_t$ can grow indefinitely with $t$, eventually reaching the limit of our precision range. Indeed, $h_t \in \{0, \ldots, t\}$, requiring $\log_2(t+1)$ bits for storage. Moreover, in practice $\mathrm{dec}$ must approximate the modulus 2 function, which is challenging to learn due to its discontinuous and periodic nature.

A more efficient solution, which implements the two-state FSA solving this problem, can still be realized by a finite precision LRNN with one layer and scalar states (and consequently also with vector states and diagonal state-transition matrices) using the recurrence $h_t = a(x_t)h_{t-1} + b(x_t)$, $h_0 = b(0) = 0$, $b(1) = a(0) = 1$, $a(1) = -1$, $y_t = h_t$. Note that the state-transition scalar $a(1)$ is negative, while current diagonal LRNNs do not allow negative values. (Sarrof et al., 2024, Theorem 2) states that this fact makes real-valued diagonal LRNNs unable to solve parity, which raises the question: *can non-diagonal LRNNs which allow only positive eigenvalues, such as DeltaNet, solve parity?* The following result answers this question negatively by generalizing Sarrof et al. (2024, Theorem 2) to non-diagonal matrices. To solve parity, the state transition matrices must allow at least one eigenvalue to be neither real nor positive. For non-diagonal matrices, this eigenvalue could simply have nonzero imaginary part. The main idea of the theorem is illustrated in Figure 2.

**Theorem 1** (Parity). *A finite precision LRNN with finitely many layers as in (1) can solve parity for arbitrary input lengths, in particular, it can recognize the language $(11)^*$, only if in at least one layer, there exist $\mathbf{x}$ such that $\mathbf{A}(\mathbf{x})$ has at least one eigenvalue $\lambda \notin \{x \in \mathbb{R} : x \geq 0\}$.*

The proof in Appendix B.1 uses the same core idea as the one in (Sarrof et al., 2024, Theorem 2). For one layer, we show that when $\mathbf{x} = 1^k$ and the conditions for the eigenvalues of $\mathbf{A}(1)$ are not met, the mapping $k \mapsto \mathbf{H}_k$ and consequently also the one $k \mapsto \hat{\mathbf{y}}_k$ will be constant (in finite precision and for large enough $k$), while $k \mapsto y_k$, with $y_k$ being the parity of $\mathbf{x}$, alternates between 0 and 1. To show this, we use the expression for the powers of the Jordan canonical form of $\mathbf{A}(1)$.

We now study the problem of counting modulo $m$, an easier version of addition modulo $m$ where the input of length $k$ never changes and is $\mathbf{x} = 1^k$, while the correct output is $y_k = (\sum_{i=1}^{k} x_i) \bmod m$. The following theorem shows that to solve this problem, products of state-transition matrices must have at least one eigenvalue with nonzero imaginary part.

**Theorem 2** (Modular Counting). *A finite precision LRNN with $L$ layers, each as in (1), can count modulo $m$, i.e. it can recognize the language $(1^m)^*$, with $m$ not a power of two, only if there exist $i \in \{1, \ldots, L\}$ and $\mathbf{x}_1, \ldots, \mathbf{x}_{2^{i-1}}$ such that for the $i$-th layer the product $\mathbf{A}(\mathbf{x}_1)\mathbf{A}(\mathbf{x}_2) \cdots \mathbf{A}(\mathbf{x}_{2^{i-1}})$ has at least one eigenvalue $\lambda$ with nonzero imaginary part, i.e. $\lambda \notin \mathbb{R}$.*

The proof is in Appendix B.2. When $L = 1$ a key step is to show that if $\mathbf{A}(1)$ has real (even negative) eigenvalues, the map $k \to \mathbf{H}_k$ will alternate between two values (in finite precision and for large enough $k$), not enough to count modulo $m > 2$. For $L > 1$, we proceed by induction using our assumption on the eigenvalues of the product of state-transition matrices.

**Discussion** Theorems 1 and 2 identify a fundamental limitation of current design choices on the structure of the state-transition matrices of LRNNs. Specifically, current LRNNs, as the ones outlined in Table 1, are incapable of solving parity, as the eigenvalues of their state-transition matrices are confined to the interval $[0, 1]$. Further, even if we allow negative eigenvalues, LRNNs using common structures for the state transition matrices, such as diagonal or triangular with real entries, cannot solve counting modulo $m$. In contrast, as we will show, LRNNs with state-transition matrices that are (products of) generalized Householder matrices, each with eigenvalues in the range $[-1, 1]$, are much more expressive.

## 4.2 ALLOWING NEGATIVE EIGENVALUES

We focus on two classes of LRNNs determined by the structure of their state-transition matrices: diagonal (such as Mamba, Mamba2, and GLA) and generalized Householder (GH, as in DeltaNet). In particular, if we let $s : \mathbb{R}^l \to [0, 1]^n$, $\phi : \mathbb{R}^l \to [0, 1]$ and $v : \mathbb{R}^l \to \mathbb{R}^n$, being learnable functions such that $\|v(x)\| = 1$ for every $x \in \mathbb{R}^l$, then the state transition matrices of each layer of many LRNNs, such as those in Table 1, can be written as either

$$A_{\text{diag}}(x) := \text{Diag}(s(x)), \quad \text{or} \quad A_{\text{GH}}(x) := I - \phi(x)v(x)v(x)^\top,$$

where $A_{\text{diag}}(x)$ is diagonal with eigenvalues $s(x)_i \in [0, 1]$, while $A_{\text{GH}}(x)$ is GH with all eigenvalues equal to one except for the one associated to the eigenvector $v(x)$, which is equal to $1 - \phi(x) \in [0, 1]$. To address the limitations discussed in the previous section, we propose the following modification that can be easily applied to LRNNs belonging to either class.

$$A_{\text{diag}}^-(x) := \text{Diag}(2s(x)-1), \quad A_{\text{GH}}^-(x) := I - 2\phi(x)v(x)v(x)^\top. \quad (2)$$

Hence, $A_{\text{diag}}^-(x)$ has eigenvalues $2s(x)_i - 1 \in [-1, 1]$ and $A_{\text{GH}}^-(x)$ has one eigenvalue equal to $1 - 2\phi(x) \in [-1, 1]$. Thus, we have extended the eigenvalues range from $[0, 1]$ to $[-1, 1]$. The norm of the matrix is still less than or equal to one, keeping the recurrence stable at long sequence lengths.

LRNNs with the modified state transition matrices can implement the solution to parity in (2) by setting $s(1) = 0$ and $\phi(1) = 1$ so that if we consider a scalar recursion, then $A_{\text{diag}}^-(1) = -1$. However, Theorem 2 shows that we cannot count modulo 3 with triangular state transition matrices, even when allowing negative eigenvalues. Therefore, in the next section, we examine the impact of our change to the eigenvalue range on non-triangular state-transition matrices.

## 4.3 EXPRESSIVITY OF PRODUCTS OF GENERALIZED HOUSEHOLDER MATRICES

We focus on state-transition matrices that are products of $k$ GH matrices. For DeltaNet $k = 1$. For any $n, k \in \mathbb{N}$, we define the set of all matrices in $\mathbb{R}^{n \times n}$ that can be expressed as a product of $k$ GH matrices, each having the only interesting eigenvalue in the range $\Omega \subseteq \mathbb{R}$, as

$$\mathcal{M}_k^n(\Omega) := \left\{ C_1 C_2 \cdots C_k \; : \; C_i = I - \beta_i v_i v_i^\top, \quad (1 - \beta_i) \in \Omega, \quad v_i \in \mathbb{R}^n, \|v_i\| = 1 \right\}. \quad (3)$$

Intuitively, higher $k$ means higher expressivity but also higher cost for matrix-vector products. Furthermore, as long as $\Omega \subseteq [-1, 1]$, the norm of the matrices is bounded by one, which guarantees that repeated matrix product do not diverge. We observe that if $M \in \mathcal{M}_1^n(\{-1\})$, then $M$ is a reflection (or Householder) matrix, and that for any $x \in \mathbb{R}^l$, $A_{\text{GH}}(x) \in \mathcal{M}_1^n([0, 1])$ and $A_{\text{GH}}^-(x) \in \mathcal{M}_1^n([-1, 1])$ so that with our change we also include reflections. Moreover, $\mathcal{M}_k^n(\Omega) \subseteq \mathcal{M}_{k'}^n(\Omega')$ if $\Omega \subseteq \Omega'$ and either $k' = k$ or $k' \geq k, 1 \in \Omega$.

Our next result shows that products of GH matrices can represent any matrix with Euclidean norm less than or equal to 1, but only when $[-1, 1] \subseteq \Omega$. In contrast, repeated products of (e.g. upper) triangular matrices with eigenvalues in $[-1, 1]$ remain triangular, with eigenvalues in the same range.

**Proposition 1** (Expressivity of products of GH matrices). *The following hold for $\mathcal{M}_k^n$ in (3):*

1. *For any $N \in \mathcal{M}_k^n([-1, 1])$, $\|N\| \leq 1$.*
2. *For any $M \in \mathbb{R}^{n \times n}$ with $\|M\| \leq 1$, then $M \in \mathcal{M}_{3n}^n([-1, 1])$ and if $M$ is orthogonal then $M \in \mathcal{M}_n^n(\{-1, 1\})$, while $M \in \mathcal{M}_{n-1}^n(\{-1, 1\})$ when $M$ is a permutation matrix.*
3. *Any eigenvalue $\lambda$ of any matrix $N \in \mathcal{M}_k^n((-1, 1])$ is either 1 or satisfies $|\lambda| < 1$ and if in addition $N \in \mathcal{M}_k^n([0, 1])$ and $k \leq 2$, then $\lambda \in [0, 1] \subset \mathbb{R}$.*

The proof in Appendix C.2 uses mainly linear algebra arguments such as the SVD decomposition and the fact that every $n \times n$ orthogonal matrix can be written as a product of $n$ reflections, due to the Cartan–Dieudonné Theorem (Gallier & Gallier, 2011).

A consequence of Proposition 1.3 is that LRNNs with layers of the form (1), where $A : \mathbb{R}^l \to \mathcal{M}_k^n([0, 1])$, have state transition matrices that are either the identity or not orthogonal, and hence cannot be reflections or rotations. Also, if $k \leq 2$ the eigenvalues are positive and hence the LRNN cannot learn parity due to Theorem 1. In contrast, if we allow $A : \mathbb{R}^l \to \mathcal{M}_k^n([-1, 1])$ and $k$ is large enough, the following theorem shows that an LRNN with one layer can implement any FSA whose transition monoid is a group, and that $n = k = 2$ is enough for cyclic groups (modular addition).

$$\begin{pmatrix} 0 & 1 & 0 \\ 1 & 0 & 0 \\ 0 & 0 & 1 \end{pmatrix} \times \begin{pmatrix} 1 & 0 & 0 \\ 0 & 0 & 1 \\ 0 & 1 & 0 \end{pmatrix} = \begin{pmatrix} 0 & 1 & 0 \\ 0 & 0 & 1 \\ 1 & 0 & 0 \end{pmatrix}$$

$$\boldsymbol{I} - 2\boldsymbol{v}_1\boldsymbol{v}_1^\top \qquad \boldsymbol{I} - 2\boldsymbol{v}_2\boldsymbol{v}_2^\top$$

Figure 3: A permutation of $k$ elements is also a composition of at most $k-1$ swaps. This maps to a product of $k-1$ Householders, each representing a swap. Illustrated for $k=3$. $\mathbf{v}_1^\top = \left(\frac{1}{\sqrt{2}}, -\frac{1}{\sqrt{2}}, 0\right), \mathbf{v}_2^\top = \left(0, \frac{1}{\sqrt{2}}, -\frac{1}{\sqrt{2}}\right)$.

**Theorem 3.** *Every FSA $\mathcal{A} = (\Sigma, Q, q_0, \delta)$ whose transition monoid $\mathcal{T}(\mathcal{A})$ is a group, can be implemented by a finite precision LRNN with one layer and $\boldsymbol{A} : \Sigma \rightarrow \mathcal{M}_{k-1}^n(\{-1, 1\})$, where $n$ is the smallest natural number such that $\mathcal{T}(\mathcal{A})$ is isomorphic to a subgroup of $S_n$, and $k = \max_{w \in \Sigma} \sum_{q \in Q} \mathbf{1}\{\delta(q, w) \neq q\}$ is the maximum number of changed states after applying a single transition. Moreover, if $\mathcal{T}(\mathcal{A})$ is isomorphic to the cyclic group $\mathbb{Z}_m$, then we can set $\boldsymbol{A} : \Sigma \rightarrow \mathcal{M}_2^2([-1, 1])$ and if $m = 2$ (parity) we can set $\boldsymbol{A} : \Sigma \rightarrow \{-1, 1\}$.*

In the proof in Appendix C.3, we map each state-transition function to a matrix representation. This can always be done using permutation matrices, but for cyclic groups, we can also use rotation matrices (Appendix C.1). For permutations, if every state-transition permutes at most $k$ states then the corresponding permutation matrix will be in $\mathcal{M}_{k-1}^n(\{-1, 1\})$, since it is either the identity or can be written as a product of at most $k - 1$ permutations of two elements (swaps), each in $\mathcal{M}_1^n(\{-1\})$ (see Figure 3). A consequence of Theorem 3 is that if every transition function of the FSA has a permutation representation corresponding to a swap or the identity, then an LRNN layer with $\boldsymbol{A} = \boldsymbol{A}_{\text{GH}}^-$, can implement it. This is useful in practice because the time complexity of an LRNN having a product of $k$ GH matrices as one state-transition matrix increases linearly with $k$. Also, for natural language tasks, the state-transitions for the FSA might be either simple or encoded using multiple letters. For example, for addition modulo 5, a word may look like "3+2+4=4" (two letters per addition). This allows an LRNN with state-transition matrices in $\mathcal{M}_1^n([-1, 1])$ to model complex transitions. Indeed, if each transition uses $k$ letters and we set $\boldsymbol{B} \equiv 0$ and $\boldsymbol{A} : \mathbb{R}^l \rightarrow \mathcal{M}_1^n([-1, 1])$ in (1), then the LRNN layer can model permutations that change up to $k + 1$ elements since

$$\boldsymbol{H}_t = \boldsymbol{C}(x_t, \dots, x_{t-k})\boldsymbol{H}_{t-k}, \quad \boldsymbol{C}(x_t, \dots, x_{t-k}) := \boldsymbol{A}(x_t)\boldsymbol{A}(x_{t-1}) \cdots \boldsymbol{A}(x_{t-k}) \in \mathcal{M}_k^n([-1, 1]).$$

In Appendix D we also show that, interestingly, an LRNN with two layers (instead of just one), each having only reflections (instead of rotations) as state-transition matrices, can solve addition modulo $m$. We now present an important result on the expressivity of LRNNs with multiple layers.

**Theorem 4.** *LRNNs with state transition matrices that are repeated products of GH matrices, each with eigenvalues in the range $[-1, 1]$, can recognize any regular language. In particular, every FSA $\mathcal{A} = (\Sigma, Q, q_0, \delta)$ can be implemented by a finite precision LRNN with $s \leq 2^{|Q|}$ layers, each of the form 1, where $n \leq |Q|$, $p \leq s$, $d = 1$, $\boldsymbol{A} : \mathbb{R}^l \rightarrow \mathcal{M}_n^n([-1, 1])$ and $\boldsymbol{B} : \mathbb{R}^l \rightarrow \mathbb{N}^n$.*

The proof in Appendix C.5 exploits the landmark Theorem by Krohn & Rhodes (1965), which states that every FSA can be decomposed as a *cascade* of simpler FSAs whose state-transition functions are either one-to-one or constant. Each layer of the LRNN will implement one FSA (with $n$ states) of the cascade using $n \times n$ permutation matrices, which are in $\mathcal{M}_{n-1}^n(\{-1, 1\})$, for the one-to-one transitions, while for constant (state-independent) transitions it will set the corresponding state-transition matrix to $0 \in \mathcal{M}_n^n(\{0\})$ and the function $\boldsymbol{B}$ appropriately. Note that we can obtain the zero matrix only inefficiently as a product of $n$ GH matrices, while it could also be obtained with a single diagonal matrix. This points towards LRNNs using a mix of GH and diagonal matrices, as recently explored by Gated DeltaNet (Yang et al., 2025) and RWKV-7.

**Discussion** The results in Theorems 3 and 4 for LRNNs are in sharp contrast with the ones for Transformers (Liu et al., 2023; Merrill & Sabharwal, 2023) and diagonal LRNNs (Merrill et al., 2024), which require either the number of layers or the precision growing with the input sequence length, and can only implement an FSA if all groups in its transition monoid are *solvable*, i.e. excluding groups isomorphic to $S_n$ with $n \geq 5$. However, compared to LRNNs without any restriction to the norm of the state-transition matrices, which need only one layer to recognize any regular language, our result requires both the number of layers and the width of the LRNN to be (in the worst case) exponential in the number of states of the FSA, although we conjecture that the number of layers might be reduced to at most linear using a more refined decomposition.

## 5 EXPERIMENTS

We investigate the effects of expanding the eigenvalue range of state-transition matrices from $[0, 1]$ to $[-1, 1]$, as explained in Section 4.2, on both synthetic tasks and language modeling. Our experiments involve Mamba, and DeltaNet, with variants trained using both the original and extended eigenvalue ranges, as shown in Table 2. We label these variants accordingly. Note that the changes increase the expressivity of Mamba and DeltaNet while coming at no additional computational cost. Detailed information on the implementation can be found in Appendix E.4.

Table 2: Summary of modifications to the state-transition matrices $\boldsymbol{A}(\boldsymbol{x}_t)$ to extend the eigenvalue range from $[0, 1]$ (Table 1) to $[-1, 1]$. We set $\boldsymbol{s}(\boldsymbol{x}_t) = \exp\left(-\boldsymbol{\Delta}_t \exp(\boldsymbol{w}_{1,i})\right)$.

|  | $[0, 1]$ | $[-1, 1]$ |
|---|---|---|
| Mamba | $\mathrm{Diag}(\boldsymbol{s}(\boldsymbol{x}_t))$ | $\mathrm{Diag}(2\boldsymbol{s}(\boldsymbol{x}_t){-}1)$ |
| DeltaNet | $\boldsymbol{I} - \beta_t \boldsymbol{k}_t \boldsymbol{k}_t^\top$ | $\boldsymbol{I} - 2\beta_t \boldsymbol{k}_t \boldsymbol{k}_t^\top$ |

### 5.1 CHOMSKY HIERARCHY

We conducted experiments with some of the formal language tasks proposed by Deletang et al. (2023) and similarly used to benchmark xLSTM (Beck et al., 2024). Our focus was on tasks where mLSTM (an LRNN) previously underperformed while sLSTM (a non-linear RNN) succeeded, specifically parity, modular arithmetic without brackets (both regular languages) and modular arithmetic with brackets (context-free language). As in Beck et al. (2024), we trained each model with sequence lengths ranging from 3 to 40 and evaluated on lengths from 40 to 256, to assess length generalization. Note that our theoretical results cover just regular languages, excluding modular arithmetic with brackets. We compared a Transformer, mLSTM and sLSTM against two variants each of Mamba and DeltaNet - with and without eigenvalue range extension.

Table 3: Performance comparison of various recurrent models on formal language tasks. We report the best of 3 runs (Table 5 in the Appendix reports the median). Scores are scaled accuracy, with 1.0 indicating perfect performance and 0.0 random guessing. The positive impact of allowing negative eigenvalues ($[-1, 1]$ range) versus restricting to positive eigenvalues ($[0, 1]$ range) is evident for both Mamba and DeltaNet. Results in parenthesis are as reported in Beck et al. (2024).

|  | **Parity** | **Mod. Arithm. (w/o brackets)** | **Mod. Arithm. (w/ brackets)** |
|---|---|---|---|
| Transformer | 0.022 | 0.031 | 0.067 |
| mLSTM | 0.087 (0.04) | 0.040 (0.04) | 0.114 (0.03) |
| sLSTM | **1.000** (1.00) | **0.787** (1.00) | **0.178** (0.57) |
| Mamba $[0, 1]$ | 0.000 | 0.095 | **0.123** |
| Mamba $[-1, 1]$ | **1.000** | **0.241** | 0.116 |
| DeltaNet $[0, 1]$ | 0.017 | 0.314 | 0.194 |
| DeltaNet $[-1, 1]$ | **1.000** | **0.971** | **0.260** |

**Results** Our findings, presented in Table 3, demonstrate that expanding the range of eigenvalues from $[0, 1]$ to $[-1, 1]$ enables all examined models to fully solve the parity task, confirming Theorem 1. For both modular arithmetic tasks, this expansion led to substantial performance improvements for Mamba and especially DeltaNet, since the latter has non-diagonal state-transition matrices that are more suited for these tasks (see Theorem 3). In Figure 6 in the Appendix, we visualize the length extrapolation performance of each model on all considered tasks. Note that we were unable to reproduce the sLSTM results reported by Beck et al. (2024) for the modular arithmetic tasks. Additional experiments and details on the tasks in Appendix E.1.

### 5.2 STATE-TRACKING

We perform experiments on group word problems, relying on the code provided by Merrill et al., 2024. We focus on the $S_5$ group—the first *unsolvable* symmetric group where current LRNNs and Transformers are known to underperform. We also report results for addition modulo 60 (i.e., the cyclic group $\mathbb{Z}_{60}$) in Appendix E.2.2, and note that parity corresponds to $S_2$. In these experiments, the model receives a sequence of group elements as input, and the supervision is another sequence of group elements, each representing the product of the preceding input elements. Since solving $S_5$ might need LRNNs with state-transition matrices formed by repeated products of four GH matrices (see Theorem 3), each with eigenvalues in $[-1, 1]$, we also consider three simplified setups: (i) allowing only permutations of up to 2 elements (identity and swaps), (ii) allowing only permutations of up to 3 elements, and (iii) using 4 tokens for each permutation. Additional details are in Ap-

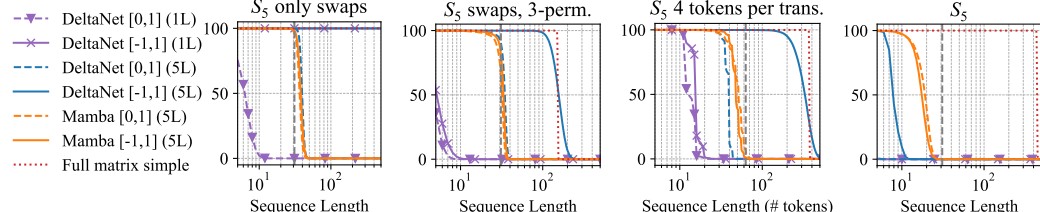

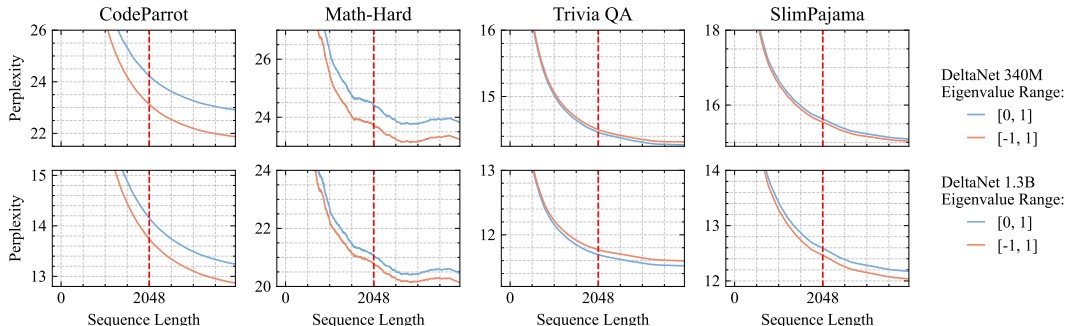

Figure 4: Sequence accuracy for varying sequence lengths on $S_5$ after 100 epochs of training. We report the best of 3 seeds for each method (in Figure 7 we report all seeds). The dashed vertical line indicates the sequence length used during training (32 except for the third plot from the left where it is 64). Each method is labeled with name, eigenvalue range, and number of layers. The dashed vertical line indicates the sequence length used during training. "Full matrix simple" is a one-layer baseline where the state update matrices are full and we have no control over the eigenvalue range.

Figure 5: Performance vs sequence length of DeltaNet variants (340M (top) and 1.3B (bottom) parameters) on four datasets. DeltaNet with eigenvalue range $[-1, 1]$ improves perplexity in coding and math compared to the $[0, 1]$ baseline. Dashed vertical line at training context length (2048).

pendix E.2. We stress that, even when restricting the inputs to only identity and swaps, the group elements for the supervision still cover the entire group, because swaps are generators of the group.

**Results** Figure 4 shows that, as predicted by Theorem 3, restricting the inputs to only swap permutations allows DeltaNet $[-1, 1]$ with even one layer to fully learn the task (since its state-transition matrices can model swaps), while DeltaNet $[0, 1]$ with 5 layers generalizes just slightly beyond the training length. In contrast, by including also permutations of 3 elements, we notice a substantial decrease in the performance of all models. Interestingly, extending the range is still advantageous in this case and DeltaNet $[-1, 1]$ with 5 layers reaches a good length generalization. Moreover, using 4 tokens per group element seems also beneficial compared to standard $S_5$, since DeltaNet $[-1, 1]$ with 5 layers manages to extrapolate very well until around length 200, which corresponds to 50 group elements, while on standard $S_5$ all models have 0 sequence accuracy prior to sequence length 30. We also report that Mamba, a diagonal LRNN, performs poorly on all setups, with and without increased eigenvalue range.

### 5.3 LANGUAGE MODELING

**Experimental Setup** We train DeltaNet models with 340M and 1.3B parameters and Mamba models with 370M parameters, each using both original and extended eigenvalue ranges. Training is done on the full FineWeb-100B dataset (Penedo et al., 2024). We chose FineWeb rather than FineWeb-Edu since it contains more code. We aligned our training pipeline with Yang et al. (2024b); see Appendix E.3.1 for details. Given our previous theoretical and experimental findings, we hypothesize that models (especially DeltaNet) with extended eigenvalue range will perform better on language modeling tasks linked to state-tracking such as coding or mathematics, compared to unmodified models. To test this hypothesis, we evaluate the perplexity of these models in a length extrapolation setup using various datasets: CodeParrot (Tunstall et al., 2022) for coding, Math-Hard (Hendrycks et al., 2021) for mathematics, TriviaQA (Joshi et al., 2017), and SlimPajama (Soboleva et al., 2023).

**Results** All models trained stably with our modification and without changing the learning rate. The validation perplexity of the proposed variants was comparable, albeit slightly worse than that of the original models throughout training (see Figure 9 in the Appendix). The experiments in Fig-

Table 4: Performance comparison using lm-harness benchmark (Gao et al., 2024) (SlimPajama (SPJ) reproduced from Yang et al. (2024b), Fine-Web (FW) ours). Results are shown for the original and extended eigenvalue range. Our models show comparable performance across tasks.

| | Model | Wiki. ppl↓ | LMB. ppl↓ | LMB. acc↑ | PIQA acc↑ | Hella. acc_n↑ | Wino. acc↑ | ARC-e acc↑ | ARC-c acc_n↑ | Avg. ↑ | SWDE cont.↑ | SQUAD cont.↑ | FDA cont.↑ |
|---|---|---|---|---|---|---|---|---|---|---|---|---|---|
| *SlimPajama 15B* | *340M params* | | | | | | | | | | | | |
| | Transformer++ | 28.39 | 42.69 | 31.0 | 63.3 | 34.0 | 50.4 | 44.5 | 24.2 | 41.2 | 42.2 | 22.1 | 21.4 |
| | Mamba [0,1] | 28.39 | 39.66 | 30.6 | 65.0 | 35.4 | 50.1 | 46.3 | 23.6 | 41.8 | 12.4 | 23.0 | 2.1 |
| | GLA [0,1] | 29.47 | 45.53 | 31.3 | 65.1 | 33.8 | 51.6 | 44.4 | 24.6 | 41.8 | 24.0 | 24.7 | 7.3 |
| | DeltaNet [0,1] | 28.24 | 37.37 | 32.1 | 64.8 | 34.3 | 52.2 | 45.8 | 23.5 | 42.1 | 26.4 | 28.9 | 12.8 |
| *FineWeb 100B* | *340M params* | | | | | | | | | | | | |
| | DeltaNet [0,1] | 24.68 | 31.49 | 33.7 | 70.3 | 45.1 | 51.3 | 50.0 | 26.1 | 46.1 | 35.2 | 28.7 | 11.8 |
| | DeltaNet [−1,1] | 24.54 | 31.15 | 34.0 | 69.9 | 44.6 | 51.9 | 50.0 | 24.4 | 45.8 | 37.2 | 33.1 | 6.6 |
| | *370M params* | | | | | | | | | | | | |
| | Mamba [0,1] | 24.84 | 24.69 | 35.6 | 70.6 | 48.4 | 51.2 | 53.4 | 24.8 | 47.3 | 21.6 | 27.7 | 2.8 |
| | Mamba [−1,1] | 25.02 | 24.71 | 36.2 | 70.5 | 47.8 | 53.3 | 54.7 | 26.7 | 48.2 | 20.9 | 24.8 | 2.5 |
| *SlimPajama 100B* | *1.3B params* | | | | | | | | | | | | |
| | Transformer++ | 16.85 | 13.44 | 48.9 | 70.8 | 49.6 | 53.6 | 56.0 | 26.5 | 50.9 | 66.6 | 31.5 | 27.4 |
| | Mamba [0,1] | 17.06 | 13.89 | 46.2 | 72.2 | 40.1 | 54.1 | 59.0 | 28.2 | 50.0 | 41.4 | 35.2 | 6.2 |
| | GLA [0,1] | 17.22 | 14.47 | 46.9 | 71.8 | 49.8 | 53.9 | 57.2 | 26.6 | 51.0 | 50.6 | 42.6 | 19.9 |
| | DeltaNet [0,1] | 16.87 | 12.21 | 48.9 | 71.2 | 50.2 | 53.6 | 57.2 | 28.3 | 51.6 | 49.5 | 37.4 | 17.2 |
| *FW 100B* | *1.3B params* | | | | | | | | | | | | |
| | DeltaNet [0,1] | 18.54 | 14.32 | 43.5 | 73.7 | 56.2 | 56.9 | 58.2 | 29.9 | 53.1 | 49.1 | 35.1 | 8.6 |
| | DeltaNet [−1,1] | 18.57 | 12.73 | 43.7 | 73.3 | 55.8 | 56.8 | 56.9 | 27.9 | 52.4 | 48.8 | 33.9 | 12.3 |

ure 5 demonstrate that on coding and math datasets, DeltaNet with an eigenvalue range of $[-1,1]$ achieves lower perplexity than the baseline with range $[0,1]$ for both model sizes. For TriviaQA, the perplexity of DeltaNet $[-1,1]$ is slightly higher. Note, that this is a task relying on memorization, not linked to state-tracking, and hence we do not expect an improvement. On SlimPajama, we also observe slight improvement with our modification. For Mamba instead, our modifications consistently degrades the performance on these tasks (Figure 10 in the Appendix).

To ensure that our models are comparable with those obtained by Yang et al. (2024b), we evaluate them on the same benchmark tasks from lm-harness (Gao et al., 2024) in Table 4. Note, that we trained on 100B tokens of FineWeb, while Yang et al. (2024b) reported results from training on 15B and 100B tokens of SlimPajama. At 340-370M parameters, with the extended range both architectures show enhanced performance in some of the tasks: Mamba in the second subset of tasks (+2.1% average accuracy) and DeltaNet in retrieval tasks (+2% SWDE, +4.4% SQUAD). At 1.3B parameters, extending the eigenvalue range of DeltaNet shows mixed results, suggesting that the increased expressivity may need training beyond 100B tokens to fully unlock the model's capacity.

# 6 CONCLUSION

In this work, we showed the substantial impact of extending the eigenvalue range of state-transition matrices in LRNNs from $[0,1]$ to $[-1,1]$. This modification provably enhances LRNN expressivity in state-tracking tasks, without adding overhead in training or inference. While Mamba successfully solves the parity problem, its diagonal matrix structure limits further gains. In contrast, DeltaNet, thanks to its non-diagonal state transition matrices which enable simultaneous token and channel mixing, excels across a broader spectrum of tasks. Our results underscore the critical role of non-diagonal state-transition matrices in augmenting state-tracking capabilities, highlighting a promising direction for future LRNN advancements.

**Limitations and Future work** Our modification is not directly compatible with a numerical technique used by some diagonal LRNNs such as Mamba2, GLA and mLSTM. In particular, these models rely on positive state-transition matrices to compute cumulative products in log space, which improves numerical accuracy and potentially training stability (see Appendix E.4 for details). Further research is needed to assess the impact of training large-scale language models with state-tracking capabilities. To this end, we aim to understand the potential downsides of increased expressivity. For example, we hypothesize a fundamental trade-off between state-tracking and associative recall, which is also of theoretical interest and could guide hybrid model design. Moreover, the theoretical expressivity of DeltaNet $[-1,1]$ with multiple layers is still unclear. We showed that it can solve addition modulo $m$ (in Appendix D) which is equivalent to the $\mathbb{Z}_3$ group word problem, but we do not know if it can also solve other word problems, such as the ones for the symmetric groups $S_n$ with $n \geq 3$.

## ACKNOWLEDGMENTS

We would like to thank David Salinas, Herilalaina Rakotoarison, Eric Alcaide, Arya Akhavan, Matia Bojovic, Erfan Mirzaei and the active members of the Flash Linear Attention discord channel for their constructive discussions and feedback. We acknowledge the support and assistance of the Data Science and Computation Facility and its Support Team, in particular Mattia Pini, in utilizing the IIT High-Performance Computing Infrastructure, on which we run our largest experiments. This research was partially supported by the following sources: PNRR MUR Project PE000013 CUP J53C22003010006 "Future Artificial Intelligence Research (FAIR)", funded by the European Union – NextGenerationEU, and EU Project ELSA under grant agreement No. 101070617. TAILOR, a project funded by EU Horizon 2020 research and innovation programme under GA No 952215; the Deutsche Forschungsgemeinschaft (DFG, German Research Foundation) under grant number 417962828; the European Research Council (ERC) Consolidator Grant "Deep Learning 2.0" (grant no. 101045765). Frank Hutter acknowledges financial support by the Hector Foundation. The authors acknowledge support from ELLIS and ELIZA. Funded by the European Union. The authors gratefully acknowledge the Gauss Center for Supercomputing eV (www.gauss-centre.eu) for funding this project by providing computing time on the GCS supercomputer JUWELS at Jülich Supercomputing Center (JSC). The MATH-HARD dataset which we use in one of our experiments was compiled from AoPS & the AoPS Community, MATHCOUNTS, the MAA, the Centre for Education in Mathematics and Computing, the Harvard-MIT Math Tournament, the Math Prize for Girls, MOEMS, the Mandelbrot Competition, and the Institute of Mathematics and Applications. Views and opinions expressed are however those of the author(s) only and do not necessarily reflect those of the European Union or the ERC. Neither the European Union nor the ERC can be held responsible for them.

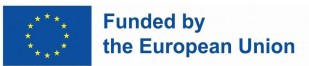

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

## SUPPLEMENTARY MATERIAL

The supplementary material is structured as follows.

- Appendix A contains additional details on the notation used, on Table 1, on the relationship between RNNs and regular languages, on the assumption of finite precision, on the states, and on the function dec.

- Appendices B and C contain the proofs for the theoretical results in Sections 4.1 and 4.3.

- Appendix D contains a theorem showing that a 2 Layer LRNN having reflections as state-transition matrices can solve addition modulo $m$.

- Appendix E contains additional details on the experiments and additonal results.

## A  ADDITIONAL BACKGROUND

### A.1  NOTATION

We denote with $\mathbb{C}, \mathbb{R}, \mathbb{N}$ the sets of complex, real, and natural numbers, respectively. We use lowercase letters for scalar quantities (e.g. $x \in \mathbb{R}$), bold lowercase letters for (column) vectors (e.g. $\boldsymbol{v} \in \mathbb{R}^n$), and bold uppercase letters for matrices (e.g. $\boldsymbol{M} \in \mathbb{R}^{n \times d}$). Some functions with matrix (vector) outputs, such as $\boldsymbol{A}$ and $\boldsymbol{B}$ in (1), are also bold upper (lower) case letters to emphasize the fact that they output matrices (vectors). We use $\odot$ to indicate the element-wise (Hadamard) product between two vectors or matrices. We denote with $\|\boldsymbol{v}\|$ the Euclidean norm of the vector $\boldsymbol{v} \in \mathbb{R}^n$. When $\boldsymbol{M} \in \mathbb{R}^{n \times d}$, $\|\boldsymbol{M}\|$ also refers to the Euclidean norm, corresponding to the largest singular value. The vector $\boldsymbol{e}_i \in \mathbb{R}^n$ is the $i$-th vector of the canonical bases in $\mathbb{R}^n$, i.e. the one-hot vector with 1 only in the $i$-th component and 0 in the others. We define the binomial coefficient for every $k, j \in \mathbb{N}$ with $j \leq k$ as

$$\binom{k}{0} := 1, \quad \binom{k}{j} := \frac{k(k-1)\ldots(k-j+1)}{j!}.$$

We also define for a Boolean $s$ and $x \in \mathbb{R}$

$$\mathbf{1}\{s\} := \begin{cases} 1 \text{ if } s \text{ is true} \\ 0 \text{ if } s \text{ is false} \end{cases}, \qquad \operatorname{sign}(x) := \begin{cases} 1 & \text{if } x \geq 0 \\ -1 & \text{if } x < 0 \end{cases}.$$

We define $\operatorname{sigmoid}(x) := 1/(1 + e^{-x})$ and $\operatorname{softplus}(x) := \ln(1 + e^x)$.

We sometimes use regular expressions (see e.g. Hopcroft & Ullman, 2001), to represent their corresponding regular language. So that e.g. $(11)^* = \{11\}^*$, where $\{11\}$ is the set containing the word 11 and $*$ is the *Kleene star* operation, is the language containing the empty word $\epsilon$ and all the words with an even number of ones, while $(1^m)^* = \{1^m\}^*$ is the language containing the words with a number of ones divisible by $m$ since $1^m$ indicates the word containing 1 repeated $m$ times. A language is *star-free* if it can be expressed with a regular expression that does not contain the Kleene star.

### A.2  DETAILS OF TABLE 1

The Mamba recurrence in Equations 3 and 4 in (Gu & Dao, 2024) is applied independently to each channel of the input sequence. Expressing the full recurrence in the matrix-form of (1) is challenging, as it would require concatenating the rows of the matrix $\boldsymbol{H}_t$. For simplicity, in Table 1 we write instead the recurrence for each row of $\boldsymbol{H}_t$. In particular, Let $\boldsymbol{x}_t \in \mathbb{R}^d$ be the input of the layer, $\boldsymbol{W}_\Delta \in \mathbb{R}^{d \times d}$, $\boldsymbol{w}_2 \in \mathbb{R}^d$, $\boldsymbol{W}_1 = (\boldsymbol{w}_1, \ldots, \boldsymbol{w}_n)^\top \in \mathbb{R}^{n \times d}$ be learnable parameters, $\boldsymbol{q}_t \in \mathbb{R}^n$, $\boldsymbol{k}_t = (k_{t,1}, \ldots, k_{t,n})^\top \in \mathbb{R}^n$ be learnable functions of the input and $\boldsymbol{\Delta}_t = \operatorname{softplus}(\boldsymbol{W}_\Delta \boldsymbol{x}_t)$. Then, if we set $\boldsymbol{H}_t = (\boldsymbol{h}_{t,1}, \ldots, \boldsymbol{h}_{t,n})^\top \in \mathbb{R}^{n \times d}$ and $\boldsymbol{H}_0 = 0$, we can write the recurrence for the $i$-th row of $\boldsymbol{H}_t$ and the output as

$$\boldsymbol{h}_{t,i} = \boldsymbol{A}_i(\boldsymbol{x}_t)\boldsymbol{h}_{t-1,i} + \boldsymbol{B}_i(\boldsymbol{x}_t), \qquad \hat{\boldsymbol{y}}_t = \psi(\boldsymbol{H}_t^\top \boldsymbol{q}_t + \boldsymbol{w}_2 \odot \boldsymbol{x}_t))$$

where $\boldsymbol{A}_i(\boldsymbol{x}_t)$ and $\boldsymbol{B}_i(\boldsymbol{x}_t)$ are the matrices stated in Table 1, i.e.

$$\boldsymbol{A}_i(\boldsymbol{x}_t) := \mathrm{Diag}\left(\exp\left(-\boldsymbol{\Delta}_t \odot \exp(\boldsymbol{w}_{1,i})\right)\right) \in \mathbb{R}^{d \times d}, \qquad \boldsymbol{B}_i(\boldsymbol{x}_t) := k_{t,i}\boldsymbol{\Delta}_t \odot \boldsymbol{x}_t \in \mathbb{R}^d.$$

Alternatively, as done in (Yang et al., 2024b, Table 4), one could write the full matrix recurrence as:

$$\boldsymbol{H}_t = \underbrace{\exp\left(-\mathbf{1}\boldsymbol{\Delta}_t^\top \odot \exp(\boldsymbol{W}_1)\right)}_{\boldsymbol{A}(\boldsymbol{x}_t)} \odot \boldsymbol{H}_{t-1} + \underbrace{\boldsymbol{k}_t(\boldsymbol{\Delta}_t \odot \boldsymbol{x}_t)^\top}_{\boldsymbol{B}(\boldsymbol{x}_t)}.$$

where $\mathbf{1}$ is the vector of $n$ ones. However, such a recurrence is not in the form (1), since we have replaced the matrix-matrix product $\boldsymbol{A}(\boldsymbol{x}_t)\boldsymbol{H}_t$ with the element-wise product $\boldsymbol{A}(\boldsymbol{x}_t) \odot \boldsymbol{H}_t$. Note that we follow the implementation of $\boldsymbol{B}(\boldsymbol{x}_t)$ used in the official Mamba codebase, which simplifies the expression originally presented in Equation 4 of (Gu & Dao, 2024) as described by the authors in a GitHub Issue[3].

## A.3 REGULAR LANGUAGES AND RECURRENT NEURAL NETWORKS

**RNNs Can Recognize Any Regular Language** A layer of a general RNN can be formulated similarly to (1) just by replacing the linear state update with a generic state-transition function $g$ as:

$$\boldsymbol{h}_t = g(\boldsymbol{h}_{t-1}, \boldsymbol{x}_t), \quad \boldsymbol{h}_0 \in \mathbb{R}^n.$$

Clearly, any FSA can be implemented by an RNN layer if $g$ is sufficiently expressive to model its state transition function.

**LRNNs Can Recognize Any Regular Language** As explained in (Liu et al., 2023, Appendix A.2) and in the proof of (Merrill et al., 2024, Theorem 5), we can implement any FSA $\mathcal{A} = (\Sigma, Q, q_0, \delta)$, and thus recognize any regular language, using matrix-vector multiplication. As a result, a single-layer LRNN by using one-hot vectors as the LRNN states and having boolean state transition matrices can recognize any language. More specifically, in (1), we can set $n = |Q|$, $\boldsymbol{H}_0 = (1, 0 \dots, 0)^\top$ and for any letter $w \in \Sigma$, $\boldsymbol{B}(w) = 0$ and $\boldsymbol{A}(w) \in \mathbb{R}^{n \times n}$ being the matrix with entries $\boldsymbol{A}(w)_{q',q} = \mathbf{1}\{\delta(w, q) = q'\}$. Note that in such a construction, the matrix $\boldsymbol{A}(w)$ can have norm greater than one, and enabling the state-transition matrix of LRNNs to have norm greater than one can make the recurrence unstable and is therefore never done in language models (see e.g. Table 1).

## A.4 FINITE PRECISION

For our positive results on LRNNs expressivity (Theorems 3 and 4), by finite precision we mean that since we have a finite number of quantities involved in the computations, then there exists a finite set $\mathbb{D} \subset \mathbb{R}$ that contains them and thus we do not require computations to be done in the reals but we can use $\mathbb{D}$ as datatype. In particular, $\mathbb{D}$ does not depend on the length of the input sequence. In practice, such data type is chosen beforehand, e.g. floating point numbers requiring a given number of bits of precision, which may not capture all quantities in our constructions.

In our negative results of Theorems 1 and 2 instead, we can pick the finite set $\mathbb{D} \subset \mathbb{R}$ arbitrarily, e.g. floating point numbers, and we also make the use of the function $\mathrm{cast} : \mathbb{R} \to \mathbb{D}$, defined in (4). that we extend to $\mathbb{C}$ by applying it separately to the real and imaginary part and to vector and matrices by applying it element-wise. The $\mathrm{cast}$ function is used because some computations of the state of the LRNN will be allowed to be in infinite precision and then transformed to finite precision using $\mathrm{cast}$ as specified in the proofs. This function provides a simplification of the actual conversion that happens in practice.

We believe that the finite precision setup is not only realistic but also allows a better focus on the drawbacks of modern LRNN. Note that for Transformers, results usually rely instead on the weaker notion of log-precision (Liu et al., 2023), meaning that the size of $\mathbb{D}$ grows logarithmically with the sequence length. This is mainly due to their limited expressivity compared to LRNNs. We also note that concerning the state-transition matrices of modern LRNNs (see Table 1), the values at the extremes of the eigenvalue range are technically not included (because of the use of the $\mathrm{sigmoid}$ and $\mathrm{softplus}$ functions). However, since we are working with finite precision, we can still include them by choosing the appropriate datatype $\mathbb{D}$, which in practice includes key values such as 0, 1, and $-1$.

---

[3] https://github.com/state-spaces/mamba/issues/19

### A.4.1 INITIAL STATE, MATRIX-VALUED STATES, AND THE DECODER FUNCTION

When introducing the LRNN layer in (1), we mention that $A$, $B$ and $\mathrm{dec}$ are learnable functions. However, to learn the constructions in our theoretical results, we need also $H_0 \subseteq \mathbb{C}^{n \times d}$ to be learnable. We do this only to simplify the results, since the same effect can also be achieved by using a special token \$ at the beginning of each sequence input to the model, called the beginning of sequence token and setting, $H_0 = 0$ for each LRNN layer so that $B(x_1)$ will have the same role as the learnable $H_0$ in our constructions. This practice is standard and used in all our experiments.

While we mention that the states $H_t$ are generally matrices of dimension $n \times d$, for our theoretical constructions (excluding the first two theorems), we set $d = 1$, so that states are vector-valued. Hence, for the problems that we consider, we find that having a matrix-valued state ($d > 1$) brings no theoretical advantage, while it is very important for associative recall.

To compute the output $\hat{y}_t$ from the state $H_t$ and the vector $x_t$ of an LRNN layer in (1), we use the function $\mathrm{dec}$, to abstract away the computations that are done on $H_t$ and $x_t$, since they are not part of the recurrence. In this work, we do not consider the internal structure of $\mathrm{dec}$, but it usually contains a normalization and a feed-forward neural network and it can approximate any continuous function.

In our negative results on LRNNs expressivity in Theorems 1 and 2, our choice of an arbitrary decoder guarantees the stronger results. For our positive results instead, we either do not consider the decoder (Theorem 3) or we make use of a linear decoder (Theorem 4). We point out that to recognize regular languages efficiently and with a smaller LRNN state it is beneficial to have a more powerful (non-linear) decoder, as in the case of word problems for cyclic or permutation groups. However, such a decoder may be hard to learn.

## B PARITY AND MODULAR COUNTING – PROOFS

We report the proofs for the theorems in Section 4.1. We start by defining the function $\mathrm{cast} : \mathbb{R} \to \mathbb{D}$, for a finite set $\mathbb{D} \subset \mathbb{R}$, which provides a simple model for the conversion of real numbers into a finite precision representation.

$$\mathrm{cast}(x) = \min_{z \in \mathcal{D}_{\min}} z, \quad \mathcal{D}_{\min} := \arg\min_{z \in \mathbb{D}} |z - x|. \tag{4}$$

Note that $\mathcal{D}_{\min}$ might not be a singleton. We naturally extend this function on complex numbers by applying it separately to the real and imaginary part, and then to complex-valued matrices by applying it element-wise. The following lemma is a key element of the proofs of Theorems 1 and 2. There, the sequence $a_k$ in the lemma takes the form of the imaginary or real part of the elements of the $k$-th power of a matrix with real eigenvalues ($\lambda_i$ will be one eigenvalue), expressed using the Jordan canonical form. See Appendix B.1 for more details on the Jordan Canonical Form. Intuitively, the lemma shows that if some of the $\lambda_i$-s are negative then for $k$ large enough, $a_k$ in finite precision will alternate between two values. Instead, if the $\lambda_i$-s are only nonnegative, $a_k$ in finite precision becomes constant for large enough $k$.

**Lemma 1.** *Let* $n, \bar{m} \in \mathbb{N}$ *and for every* $k > \bar{m}$ *let*

$$a_k := \sum_{i=1}^{n} c_i \binom{k}{m_i} \lambda_i^{k - m_i}, \quad \text{with } c_i, \lambda_i \in \mathbb{R}, m_i \in \mathbb{N}, m_i \leq \bar{m}, \quad \forall i \in \{1, \dots, n\},$$

*then there exist* $\bar{k} \in \mathbb{N}$ *such that for every* $k \geq \bar{k}$ *there exist* $\bar{a}_1, \bar{a}_2 \in \mathbb{D}$ *such that*

$$\mathrm{cast}(a_{2k}) = \bar{a}_1, \quad \mathrm{cast}(a_{2k+1}) = \bar{a}_2.$$

*Furthermore, if* $\lambda_i \geq 0$ *for every* $i \in \{1, \dots, n\}$*, then* $\mathrm{cast}(a_k) = \bar{a}_1 = \bar{a}_2$ *for* $k \geq \bar{k}$*.*

*Proof.* If $c_i = 0$ for every $i$, or $\lambda_i = 0$ for every $i$, then $a_k = 0$ for all $k$ and the statement is trivially satisfied. Without loss of generality we can assume that that $c_i \neq 0$ and $\lambda_i \neq 0$ for every $i \in \{1, \dots, n\}$, since for each $i$ where this is not true we can remove the corresponding term in the sum (since it will be 0) and use smaller value for $n$. We divide the proof into two parts.

**Positive powers:** Assume that $\lambda_i > 0$ for all $i \in \{1, \dots, n\}$. This yields that for every $i$ and every $k > \bar{m}$, $\binom{k}{m_i} \lambda_i^{k - m_i} > 0$. Since the $\mathrm{cast}$ function is piecewise constant with a finite number of pieces,

we can divide the real line into a finite number of intervals where cast is constant. We now show that for $k$ large enough, the interval where $a_k$ belongs, and hence $\mathrm{cast}(a_k)$, does not vary with $k$.

Without loss of generality we assume that for every $i, j \in \{1, \ldots, n\}$ we have that $(m_i, \lambda_i) \neq (m_j, \lambda_j)$, since otherwise we can factor out $\binom{k}{m_i}\lambda_i^{k-m_i}$ and use a smaller $n$. Note that $\binom{k}{m_i}\lambda_i^{k-m_i} = \frac{k(k-1)\cdots(k-m_i+1)}{m_i!}\lambda_i^{k-m_i}$ and hence $g_i(k) = \binom{k}{m_i}\lambda_i^{k-m_i}$ for large $k$ behaves like the function $k^{m_i}\lambda^k$, i.e. the product of a polynomial and an exponential function of $k$. Without loss of generality, we therefore take the order of the indices of the terms in the sum such that the functions $g_i$ are in decreasing order of growth:

$$\lambda_i > \lambda_j \text{ or } \lambda_i = \lambda_j, m_i > m_j \qquad \forall i, j : i > j.$$

By factoring out $g_1(k)$, i.e. the fastest growing term, from $a_k$ we get

$$a_k = \binom{k}{m_1}\lambda_1^{k-m_1}(c_1 + b_k) \qquad b_k := \sum_{i=2}^{n} c_i \frac{\binom{k}{m_i}\lambda_i^{k-m_i}}{\binom{k}{m_1}\lambda_1^{k-m_1}},$$

with $\lim_{k\to\infty} b_k = 0$ and therefore, since for every $i$ and every $k > \bar{m}$, $\binom{k}{m_1}\lambda_1^{k-m_i} > 0$ and $c_1 \neq 0$, there exist $\hat{k} \in \mathbb{N}$ such that for every $k \geq \hat{k}$, $\mathrm{sign}(a_k) = \mathrm{sign}(c_1 + b_k) = \mathrm{sign}(c_1)$. Now let $\mathbb{D} = \{z_1, \ldots, z_d\}$ with $z_1 < z_2 < \cdots < z_d$ and let $y_1 = -\infty$, $y_{d+1} = \infty$ and $y_i = (z_{i-1} + z_i)/2$ for $i \in \{2, \ldots, d\}$. From its definition, cast is a piecewise constant function such that $\mathrm{cast}(x) = z_i$ for every $x \in (y_i, y_{i+1})$. We now consider three cases according to the values of $\lambda_1$ and $m_1$.

**1)** If $\lambda_1 > 1$ or $\lambda_1 = 1, m_1 > 0$, then $\lim_{k\to\infty}\binom{k}{m_1}\lambda_1^{k-m_i} = \infty$ and there exists $\bar{k} \geq \hat{k}$ such that for every $k \geq \bar{k}$, either $a_k > y_d$ (if $\mathrm{sign}(c_1) = 1$) or $a_k < y_2$ (if $\mathrm{sign}(c_1) = -1$) and hence $\mathrm{cast}(a_k) = \bar{a} \in \{z_1, z_d\}$.

**2)** If $\lambda_1 < 1$ then $\lim_{k\to\infty}\binom{k}{m_1}\lambda_1^{k-m_i} = 0$ and hence there exist $\epsilon > 0$, $j \in \{1, \ldots, d\}$, $\bar{k} > \hat{k}$ such that for every $k \geq \bar{k}$, $a_k \in \Omega \subseteq (y_j, y_{j+1})$, where $\Omega = (0, \epsilon)$ if $\mathrm{sign}(c_1) = 1$ and $\Omega = (-\epsilon, 0)$ if $\mathrm{sign}(c_1) = -1$. Therefore, $\mathrm{cast}(a_k) = z_j$ for every $k \geq \bar{k}$.

**3)** If $\lambda_1 = 1, m_1 = 0$, then $\binom{k}{m_1}\lambda_1^{k-m_i} = 1$ for every $k$ and hence

$$a_k = c_1 + b_k, \quad b_k = \sum_{i=2}^{n} c_i \binom{k}{m_i}\lambda_i^{k-m_i} \qquad \text{with } \lambda_i < 1 \ \forall i \in \{2, \ldots, n\}$$

Note that $b_k$ has now the same structure as $a_k$, just with one less term in the sum, therefore we can factor out the term $\binom{\lambda_2}{m_2}\lambda^{k-m_2}$ and, since $\lambda_2 < 1$, apply the same reasoning as for the second case ($\lambda_1 < 1$) to $c_1 + b_k$ and prove that there exist $\epsilon > 0$, $j \in \{1, \ldots, d\}$, $\bar{k} > \hat{k}$ such that for every $k \geq \bar{k}$, we have that $\mathrm{sign}(b_k) = \mathrm{sign}(c_2)$, $a_k \in \Omega \subseteq (y_j, y_{j+1})$, where $\Omega = (c_1, \epsilon)$ if $\mathrm{sign}(c_2) = 1$ and $\Omega = (-\epsilon, c_1)$ if $\mathrm{sign}(c_2) = -1$. Therefore $\mathrm{cast}(a_k) = z_j$ for every $k \geq \bar{k}$.

In summary, we proved that when $\lambda_i \geq 0$ for every $i$, there exist $\bar{a} \in \mathbb{D}$, $\bar{k} \in \mathbb{N}$ such that for every $k \geq \bar{k}$ $a_k = \bar{a}$, which concludes the first part of the proof.

**Some powers can be negative:** Consider the general case where $\lambda_i \in \mathbb{R}$ can be negative. We can write

$$a_k = \sum_{i=1}^{n} c_i \binom{k}{m_i}\mathrm{sign}(\lambda_i)^{k-m_i}|\lambda_i|^{k-m_i}.$$

Since $\mathrm{sign}(x)^{2k-m_i}$ and $\mathrm{sign}(x)^{2k+1-m_i}$ do not vary with $k$ we consider the two subsequences

$$a_{2k} = \sum_{i=1}^{n} \hat{c}_i \binom{2k}{m_i}|\lambda_i|^{2k-m_i}, \quad \hat{c}_i = c_i\mathrm{sign}(\lambda_i)^{2k-m_i}$$

$$a_{2k+1} = \sum_{i=1}^{n} \tilde{c}_i \binom{2k+1}{m_i}|\lambda_i|^{2k+1-m_i}, \quad \tilde{c}_i = c_i\mathrm{sign}(\lambda_i)^{2k+1-m_i},$$

and we can apply the same proof as for the case when $\lambda_i > 0$ for every $i$ to each of the subsequences above, which gives the final result in the case $\lambda_i \in \mathbb{R}$ for every $i$. $\qquad \square$

### B.1 PROOF OF THEOREM 1

The language $(11)^*$ contains all sequences with an even number of ones. An FSA recognizing the language, for the sequence $1^k$ will output $y_k = 1$ if $k$ is even and $y_k = 0$ if $k$ is odd. Consider an LRNN with one layer as in (1). We will prove that if $A(1)$ has only nonnegative eigenvalues, then there exists a $\bar{k} > 0$ such that for every $k \geq \bar{k}$, the finite precision version of the state $H_k$ corresponding to the sequence $1^k$ does not depend on $k$ and is equal to $\overline{H}$. Hence, no matter the choice of dec, also the finite precision version of $\hat{y}_k$ will not vary with $k$ and thus for some $k' \geq \bar{k}$, $\hat{y}_{k'} \neq k' \mod 2 = y_{k'}$. An inductive argument can then be used for the case of LRNNs with multiple (finitely many) layers, using the fact that the input of the next layer will be constant for $k$ large enough, as the input of the first layers.

By unrolling the recursion in 1 we obtain a closed-form expression for the state

$$H_k = \sum_{i=1}^{k-1} \left( \prod_{j=i+1}^{k-1} A(x_j) \right) B(x_i) + \left( \prod_{i=1}^{k} A(x_i) \right) H_0,$$

where we set $\prod_{j=k}^{k-1} A(x_j) = I$ to avoid clutter. We follow Merrill et al. (2024) and make the simplifying assumption that in finite precision the state at time $k$ is computed by first evaluating all products involving the matrices $A(x_j)$ separately and in infinite precision, followed by casting them into finite precision, and finally executing the sum also in infinite precision and casting the result in finite precision. This avoids having to deal with the individual matrix sums and products in finite precision, which would break associativity and be harder to analyze. Hence, if we set $x_1 \ldots x_k = 1^k$, we get the following exact and finite precision expressions for the state at time $k$.

$$H_k = \sum_{i=0}^{k-1} A(1)^i B(1) + A(1)^k H_0, \quad \widehat{H}_k = \mathrm{cast}\left( \sum_{i=0}^{k-1} \mathrm{cast}\left( A(1)^i B(1) \right) + \mathrm{cast}\left( A(1)^k H_0 \right) \right),$$

where cast, defined in (4), is an operation that converts matrices with complex values element-wise into finite precision by e.g. separately converting real and imaginary parts.

Using the Jordan canonical form theorem (see e.g. Horn & Johnson, 2012, Chap. 3.1), we can write $A(1) = PJP^{-1}$, where $J$ is block diagonal made of the Jordan blocks $J_1, \ldots, J_s$ with $s \leq n$, $J_i \in \mathbb{R}^{k_i \times k_i}$ and with corresponding complex eigenvalues $\lambda_1 \ldots \lambda_s$ (with multiplicity taken into account). Such decomposition is useful because it allows, for $k \geq \max_i k_i - 1$, to write

$$A(1)^k = PJ^k P^{-1}, \quad J_i^k = \begin{bmatrix} \lambda_i^k & \binom{k}{1}\lambda_i^{k-1} & \binom{k}{2}\lambda_i^{k-2} & \cdots & \cdots & \binom{k}{k_i-1}\lambda_i^{k-k_i+1} \\ & \lambda_i^k & \binom{k}{1}\lambda_i^{k-1} & \cdots & \cdots & \binom{k}{k_i-2}\lambda_i^{k-k_i+2} \\ & & \ddots & \ddots & \ddots & \vdots & \vdots \\ & & & \ddots & \ddots & \vdots \\ & & & & \lambda_i^k & \binom{k}{1}\lambda_i^{k-1} \\ & & & & & \lambda_i^k \end{bmatrix}.$$

Then, from the structure of the Jordan decomposition, the imaginary and real part of each element of the matrices $A(1)^k B(1)$ and $A(1)^k H_0$ will be a linear combination of elements of the Jordan blocks taking the same form of $a_k$ in Lemma 1. Therefore since $\lambda_i \geq 0$ for every $i$, we can apply Lemma 1 component-wise and conclude that there exists $\tau \in \mathbb{N}$, $\widehat{C} \in \mathbb{C}^{n \times d}$ and $\widehat{D} \in \mathbb{C}^{n \times d}$ such that for every $k \geq \tau$, $\widehat{C}_k = \mathrm{cast}(A(1)^k B(1)) = \widehat{C}$ and $\widehat{D}_k = \mathrm{cast}(A(1)^k H_0) = \widehat{D}$ and hence

$$\widehat{H}_k = \mathrm{cast}\left( \sum_{i=0}^{\tau-1} \widehat{C}_i + \widehat{D} + (1-\tau)\widehat{C} + k\widehat{C} \right).$$

Note that only the matrix $k\widehat{C}$ varies with $k$ and for large enough $k$, the real and imaginary parts of each element of $k\widehat{C}$ will be either 0, smaller than $\min_{x \in \mathbb{R}} \mathrm{cast}(x)$ or larger than $\max_{x \in \mathbb{R}} \mathrm{cast}(x)$. Therefore, we obtain that there exists $\overline{H} \in \mathbb{C}^{n \times d}$ and $\bar{k} \geq \tau$ such that for every $k \geq \bar{k}$ we have $\widehat{H}_k = \overline{H}$, which concludes the proof. $\square$

## B.2  PROOF OF THEOREM 2

**One Layer** Let $\widehat{\boldsymbol{H}}_k$ and $\hat{y}_k := \mathrm{cast}(\mathrm{dec}(\widehat{\boldsymbol{H}}_k, x_k))$ be the finite precision versions of the state $\boldsymbol{H}_k$ and (scalar) output of a one-layer LRNN on the input $\boldsymbol{x} = x_1 \ldots x_k = 1^k$. Let also $y_k = \mathbf{1}\{k \bmod m = 0\}$ be the correct output recognizing the word $\boldsymbol{x}$. We will show that if the assumptions on the eigenvalues are not satisfied, i.e. if for any $x$, every eigenvalue $\lambda$ of $\boldsymbol{A}(x)$ is real, then there exist $\overline{\boldsymbol{H}}_1, \overline{\boldsymbol{H}}_2 \in \mathbb{C}^{n \times n}$, $\bar{y}_1, \bar{y}_2 \in \mathbb{R}^p$ and $\tau \in \mathbb{N}$ such that for all $k \geq \tau$

$$\widehat{\boldsymbol{H}}_k := \begin{cases} \overline{\boldsymbol{H}}_1 & \text{if } k \bmod 2 = 0 \\ \overline{\boldsymbol{H}}_2 & \text{otherwise} \end{cases} \quad , \quad \hat{y}_k = \begin{cases} \bar{y}_1 & \text{if } k \bmod 2 = 0 \\ \bar{y}_2 & \text{otherwise} \end{cases} \tag{5}$$

where without loss of generality we take $\bar{y}_1, \bar{y}_2 \in \{0, 1\}$. If $\bar{y}_1 = \bar{y}_2$, then, similarly to parity, $\hat{y}_k = \hat{y}_{k+1}$ for all $k > \tau$, while since $m > 2$, if $k \bmod m = m - 1$, then $1 = y_{k+1} \neq y_k = 0$. Otherwise if $\bar{y}_1 \neq \bar{y}_2$ then if we assume that $k \bmod d = 1$ and $\hat{y}_k = y_k = 0$, then $1 = \hat{y}_{k+1} \neq y_{k+1} = 0$ since $m > 2$. This will prove the result for a one-layer LRNN. Then, we will proceed with the proof of finitely many layers.

To prove (5), we set

$$\widehat{\boldsymbol{H}}_k = \mathrm{cast}\left(\sum_{i=0}^{k-1} \mathrm{cast}\left(\boldsymbol{A}(1)^i \boldsymbol{B}(1)\right) + \mathrm{cast}\left(\boldsymbol{A}(1)^k \boldsymbol{H}_0\right)\right),$$

and proceed similarly to Theorem 1. Indeed, using the $k$-th power formula for the Jordan Decomposition of the matrix $\boldsymbol{A}(1)$ with eigenvalues $\lambda_1, \ldots, \lambda_s$, the imaginary and real part of each element of the matrices $\boldsymbol{A}(1)^k \boldsymbol{B}(1)$ and $\boldsymbol{A}(1)^k \boldsymbol{H}_0$ will be a linear combination of elements of the Jordan blocks taking the same form of $a_k$ in Lemma 1. Therefore since our assumptions with $L = 1$ imply that $\lambda_i \in \mathbb{R}$ for every $i$, we can apply Lemma 1 to show that there exist $\bar{\tau} \in \mathbb{N}$, $\overline{\boldsymbol{C}}_1, \overline{\boldsymbol{C}}_2, \overline{\boldsymbol{D}}_1, \overline{\boldsymbol{D}}_2 \in \mathbb{C}^{n \times d}$ such that for every $k \geq \tau$ we have

$$\widehat{\boldsymbol{C}}_k := \mathrm{cast}(\boldsymbol{A}(1)^k \boldsymbol{B}) = \begin{cases} \overline{\boldsymbol{C}}_1 & \text{if } k \bmod 2 = 1 \\ \overline{\boldsymbol{C}}_2 & \text{if } k \bmod 2 = 0 \end{cases} \quad \widehat{\boldsymbol{D}}_k := \mathrm{cast}(\boldsymbol{A}(1)^k \boldsymbol{H}_0) = \begin{cases} \overline{\boldsymbol{D}}_1 & \text{if } k \bmod 2 = 1 \\ \overline{\boldsymbol{D}}_2 & \text{if } k \bmod 2 = 0 \end{cases}$$

Finally, if for simplicity we consider $\tau \bmod 2 = 0$, we have that for $2k \geq \tau$

$$\widehat{\boldsymbol{H}}_{2k} = \mathrm{cast}\left(\sum_{i=1}^{\tau-1} \widehat{\boldsymbol{C}}_i + \left(k - \frac{\tau}{2} + 1\right)\overline{\boldsymbol{C}}_2 + \left(k - \frac{\tau}{2}\right)\overline{\boldsymbol{C}}_1 + k\overline{\boldsymbol{D}}_2\right)$$

$$\widehat{\boldsymbol{H}}_{2k+1} = \mathrm{cast}\left(\sum_{i=1}^{\tau-1} \widehat{\boldsymbol{C}}_i + \left(k - \frac{\tau}{2} + 1\right)(\overline{\boldsymbol{C}}_2 + \overline{\boldsymbol{C}}_1) + k\overline{\boldsymbol{D}}_1\right)$$

where by factoring out $k$ inside cast, we note that for large enough $k$, the real and imaginary parts of each element of the matrices inside cast will be either constant, smaller than $\min_{x \in \mathbb{R}} \mathrm{cast}(x)$ or larger than $\max_{x \in \mathbb{R}} \mathrm{cast}(x)$. Thus there exist $\overline{\boldsymbol{H}}_1, \overline{\boldsymbol{H}}_2 \in \mathbb{C}^{n \times d}$ and $\bar{k} \geq \tau$ such that (5) is satisfied, concluding the proof for the case of a single layer.

**Multiple Layers** Note that for one layer we have two subsequences (one of even and one of odd elements) of the output sequence $\hat{\boldsymbol{y}}_1, \hat{\boldsymbol{y}}_2, \ldots$ converging after a finite number of elements. This means that there exist $\boldsymbol{a}, \boldsymbol{b} \in \mathbb{R}^p$ such that for all $k \geq \bar{k}$ we have

$$\hat{\boldsymbol{y}}_{2k} = \boldsymbol{a}, \quad \hat{\boldsymbol{y}}_{2k+1} = \boldsymbol{b}.$$

Now, consider an additional layer that takes as input $\boldsymbol{x}_1^{(2)}, \ldots, \boldsymbol{x}_k^{(2)}$, with $\boldsymbol{x}_i^{(2)} = \hat{\boldsymbol{y}}_i$ and outputs $\hat{\boldsymbol{y}}_1^{(2)}, \ldots, \hat{\boldsymbol{y}}_k^{(2)}$ as

$$\boldsymbol{H}_k^{(2)} = \boldsymbol{A}^{(2)}(\boldsymbol{x}_k^{(2)})\boldsymbol{H}_{k-1}^{(2)} + \boldsymbol{B}^{(2)}(\boldsymbol{x}_k^{(2)}), \quad \hat{\boldsymbol{y}}_k^{(2)} = \mathrm{dec}^{(2)}(\boldsymbol{H}_k^{(2)}, \boldsymbol{x}_k^{(2)}).$$

Without loss of generality, assume for simplicity that $\bar{k} = 1$ and that $\hat{\boldsymbol{x}}_{2k}^{(2)} = \boldsymbol{a}$ and $\hat{\boldsymbol{x}}_{2k+1}^{(2)} = \boldsymbol{b}$ for all $k$. If we set

$$\boldsymbol{A}_1 := \boldsymbol{A}^{(2)}(\boldsymbol{a}), \qquad \boldsymbol{A}_2 := \boldsymbol{A}^{(2)}(\boldsymbol{b}),$$
$$\boldsymbol{B}_1 := \boldsymbol{B}^{(2)}(\boldsymbol{a}), \qquad \boldsymbol{B}_2 := \boldsymbol{B}^{(2)}(\boldsymbol{b}),$$
$$\boldsymbol{C}_1 := \boldsymbol{A}_1 \boldsymbol{A}_2, \qquad \boldsymbol{C}_2 := \boldsymbol{A}_1 \boldsymbol{B}_2 + \boldsymbol{B}_1,$$

then we can write the states of the second layer at even indices as

$$\boldsymbol{H}_{2k}^{(2)} = \boldsymbol{A}_1 \boldsymbol{H}_{2k-1}^{(2)} + \boldsymbol{B}_1 = \boldsymbol{A}_1 \boldsymbol{A}_2 \boldsymbol{H}_{2k-2}^{(2)} + \boldsymbol{A}_1 \boldsymbol{B}_2 + \boldsymbol{B}_1$$

$$= \boldsymbol{C}_1 \boldsymbol{H}_{2(k-1)}^{(2)} + \boldsymbol{C}_2 = \sum_{i=0}^{k-1} \boldsymbol{C}_1^i \boldsymbol{C}_2 + \boldsymbol{C}_1^k \boldsymbol{H}_0$$

Furthermore, for the states at odd indices, we have

$$\boldsymbol{H}_{2k+1}^{(2)} = \boldsymbol{A}_2 \boldsymbol{H}_{2k}^{(2)} + \boldsymbol{B}_2 = \sum_{i=0}^{k-1} \boldsymbol{A}_2 \boldsymbol{C}_1^i \boldsymbol{C}_2 + \boldsymbol{A}_2 \boldsymbol{C}_1^k \boldsymbol{H}_0 + \boldsymbol{B}_2.$$

We notice that the sequences $\boldsymbol{H}_{2k}^{(2)}$ and $\boldsymbol{H}_{2k+1}^{(2)}$ are in a form similar to $\boldsymbol{H}_k$ of the first layer. If the assumption on the eigenvalues of the state-transition matrices of the second layer does not hold, this means that for all $\boldsymbol{x}, \boldsymbol{y}$ each eigenvalue of $\boldsymbol{A}^{(2)}(\boldsymbol{x})\boldsymbol{A}^{(2)}(\boldsymbol{y})$, including $\boldsymbol{C}_1$, is real (but possibly negative). Therefore, we can proceed similarly to the case of one layer, i.e. using the powers of the Jordan canonical form of $\boldsymbol{C}_1$, to show that if we let $\widehat{\boldsymbol{H}}_{2k}^{(2)}$ and $\widehat{\boldsymbol{H}}_{2k+1}^{(2)}$ being the finite precision counterparts of $\boldsymbol{H}_{2k}^{(2)}$ and $\boldsymbol{H}_{2k+1}^{(2)}$, then there exist $\overline{\boldsymbol{H}}_1^{(2)}, \overline{\boldsymbol{H}}_2^{(2)}, \overline{\boldsymbol{H}}_3^{(2)}, \overline{\boldsymbol{H}}_4^{(2)} \in \mathbb{C}^{n \times d}, \bar{k}_2 \geq 0$ such that for every $k \geq \bar{k}$

$$\widehat{\boldsymbol{H}}_{2k}^{(2)} = \begin{cases} \overline{\boldsymbol{H}}_1^{(2)} & \text{if } k \bmod 2 = 0 \\ \overline{\boldsymbol{H}}_2^{(2)} & \text{if } k \bmod 2 = 1 \end{cases} , \quad \widehat{\boldsymbol{H}}_{2k+1}^{(2)} = \begin{cases} \overline{\boldsymbol{H}}_3^{(2)} & \text{if } k \bmod 2 = 0 \\ \overline{\boldsymbol{H}}_4^{(2)} & \text{if } k \bmod 2 = 1 \end{cases} .$$

Therefore, for $k \geq \bar{k}_2$, the function $k \mapsto \overline{\boldsymbol{H}}_k^{(2)}$ will be periodic with period a divisor of four and hence no matter the choice of $\text{dec}^{(2)}$, also the function $k \mapsto \hat{\boldsymbol{y}}_k^{(2)}$ will be periodic with period a divisor of 4. Consequently, with two layers one can recognize the language $(1^m)^*$ only when $m = 1$, $m = 2$, or $m = 4$, since those are the only cases where $k \mapsto y_k$ has a period which is a divisor of 4. Thanks to the assumption on the eigenvalues of the products of state-transition matrices, we can extend this argument inductively to the case of an LRNN with $L$ layers. In particular, for the $i$-th layer, the induction hypothesis is that we assume $k \mapsto \boldsymbol{x}_k^{(i)}$, mapping $k$ to the $k$-th input to the layer, to be periodic with period a divisor of $2^{i-1}$ for $k$ large enough. Hence, there will be $2^{i-1}$ subsequences of states, each containing powers of the product of $2^{i-1}$ state-transition matrices. From our hypothesis on the eigenvalues of products of state-transition matrices, such product will have only real eigenvalues and hence each subsequence will have 2 converging subsequences resulting in $k \mapsto \boldsymbol{H}_k^{(i)}$ and consequently $k \mapsto \hat{\boldsymbol{y}}_k^{(i)}$ and hence $k \mapsto \boldsymbol{x}_k^{(i+1)}$, for $k$ large enough, being periodic with period a divisor of $2^i$. Therefore, for the $L$-th layer, there exists $\bar{k}_L \geq 0$ such that for every $k \geq \bar{k}_L$, the function $k \mapsto \hat{\boldsymbol{y}}_k^{(L)}$ is periodic with a period which is a divisor of $2^L$ and thus it can recognize the language $(1^m)^*$ only when $2^L \bmod m = 0$, which happens only when there exists $p \leq L$ such that $m = 2^p$ and hence $m$ is a power of two, ending the proof. $\square$

## C  PRODUCTS OF GENERALIZED HOUSEHOLDER MATRICES – PROOFS

We provide proofs for the results stated in Section 4.3. Before that, we illustrate how a linear RNN with one layer and state transition matrices that are products of 2 Householder matrices can count modulo $m$.

### C.1  PRODUCTS OF TWO HOUSEHOLDERS AND MODULAR COUNTING

Counting modulo $m$ can be achieved by rotating a vector in $\mathbb{R}^2$ by an angle of $2\pi/m$ radians, and we can express a rotation matrix as a product of two reflection matrices, which are GH matrices with eigenvalues in $\{-1, 1\}$ (see Appendix C.1). Inded, for any $m \in \mathbb{N}$ there exist unit norm vectors $\boldsymbol{v}_1, \boldsymbol{v}_2 \in \mathbb{R}^2$ such that

$$\boldsymbol{R}(\theta) := \begin{bmatrix} \cos\theta & -\sin\theta \\ \sin\theta & \cos\theta \end{bmatrix} = \left(\boldsymbol{I} - 2\boldsymbol{v}_1\boldsymbol{v}_1^\top\right)\left(\boldsymbol{I} - 2\boldsymbol{v}_2\boldsymbol{v}_2^\top\right), \quad \theta = \frac{2\pi}{m}.$$

If we set the state-transition matrix in (1) to $\boldsymbol{A}(1) = \boldsymbol{R}(\theta)$, an LRNN with one layer can count modulo $m$, since if we also set $\boldsymbol{H}_0 = (1,0)^\top$ and $\dec(\boldsymbol{H}, x) = \arg\max_i \boldsymbol{D}_i^\top \boldsymbol{H}$, with $\boldsymbol{D}_i = \boldsymbol{R}(i\theta)\boldsymbol{H}_0$ for all $i \in \{0, \dots, m-1\}$, then for the input $\boldsymbol{x} = 1^t$ and since $\boldsymbol{R}$ has period $2\pi$, we get

$$\hat{y}_t = \dec(\boldsymbol{H}_t, 1) = \dec(\boldsymbol{A}(1)^t \boldsymbol{H}_0, 1) = \dec(\boldsymbol{R}(t\theta)\boldsymbol{H}_0, 1) = t \bmod m.$$

## C.2 Proof of Proposition 1

**First item** It can be shown by noting that if $\boldsymbol{C} \in \mathcal{M}_1^n([-1,1])$, then $\|\boldsymbol{C}\| \leq 1$ and using the sub-multiplicative property of the Euclidean norm, i.e the fact that $\|\boldsymbol{A}\boldsymbol{B}\| \leq \|\boldsymbol{A}\|\|\boldsymbol{B}\|$.

**Second item** Note that any real matrix has a singular value decomposition. Hence we can write

$$\boldsymbol{M} = \boldsymbol{U}\boldsymbol{S}\boldsymbol{V}^\top$$

with $\boldsymbol{U}, \boldsymbol{V} \in \mathbb{R}^{n\times n}$ orthogonal and $\boldsymbol{S} = \mathrm{Diag}(\sigma_1, \dots, \sigma_n)$ with $\sigma_i \in [0,1]$, since $\|\boldsymbol{M}\| \leq 1$. It follows from the $n$-reflections theorem[4] that we can write $\boldsymbol{U}$ and $\boldsymbol{V}$ as either the identity $\boldsymbol{I} \in \mathcal{M}_1^n(\{1\})$ or the product of at most $n$ reflections, each of which is in $\mathcal{M}_1^n(\{-1\})$. Hence $\boldsymbol{U}, \boldsymbol{V} \in \mathcal{M}_n^n(\{-1,1\})$. We can also write the matrix $\boldsymbol{S}$ as the product of $n$ GH matrices as

$$\boldsymbol{S} = \boldsymbol{S}_1 \boldsymbol{S}_2 \dots \boldsymbol{S}_n, \quad \boldsymbol{S}_i = \boldsymbol{I} - (1-\sigma_i)\boldsymbol{e}_i \boldsymbol{e}_i^\top$$

where $\boldsymbol{e}_i$ is the $i$-th element of the canonical basis of $\mathbb{R}^n$. Hence, $\boldsymbol{S} \in \mathcal{M}_n^n([0,1])$. The proof of the first part is concluded since we wrote each of $\boldsymbol{U}, \boldsymbol{S}, \boldsymbol{V}$ as a product of at most $n$ GH matrices. If $\boldsymbol{M}$ is orthogonal, we apply the $n$-reflections theorem directly. We also note that if $\boldsymbol{M} = \boldsymbol{P} \in \{0,1\}^{n\times n}$ with $\boldsymbol{P}$ being a permutation matrix different from the identity, it can be written as products of at most $n-1$ *swaps*, i.e. permutation matrices permuting only two elements. Therefore we have that there exists an integer $k \leq n-1$ and indices $i_1, \dots, i_k$ and $j_1, \dots, j_k$ such that $i_l \neq j_l$ and

$$\boldsymbol{P} = \prod_{l=1}^{k-1} \boldsymbol{P}_{i_l j_l}, \quad \boldsymbol{P}_{ij} = (\boldsymbol{I} - 2\boldsymbol{v}_{ij}\boldsymbol{v}_{ij}^\top) \quad v_{ijl} = \begin{cases} 1/\sqrt{2} & \text{if } l=i \\ -1/\sqrt{2} & \text{if } l=j \\ 0 & \text{otherwise} \end{cases},$$

where we set $\boldsymbol{v}_{ij} = (v_{ij1}, \dots, v_{ijn})$. Note that since $\|\boldsymbol{v}_{ij}\| = 1$, $\boldsymbol{P}_{ij} \in \mathcal{M}_k^n(\{-1\})$ with $k \leq n$. For the the case where $\boldsymbol{M} = \boldsymbol{I}$ we can use the fact that $\boldsymbol{I} \in \mathcal{M}_1^n(\{1\})$.

**Third item** Let $\boldsymbol{N} = \boldsymbol{C}_1 \boldsymbol{C}_2 \cdots \boldsymbol{C}_k \in \mathcal{M}_k^n((-1,1])$, with $\boldsymbol{C}_i = \boldsymbol{I} - \beta_i \boldsymbol{z}_i \boldsymbol{z}_i^\top$ with $\|\boldsymbol{z}_i\| = 1$ and $\beta_i \in [0,2)$. If $\boldsymbol{N} = \boldsymbol{I}$ the statement is satisfied, otherwise, let $\mathcal{V} = \mathrm{span}\{\boldsymbol{z}_i : i \in \{1, \dots, k\}, \beta_i > 0\}$. Any unit vector $\boldsymbol{v} \in \mathbb{R}^n$ can then be written as $\boldsymbol{v} = \boldsymbol{v}_1 + \boldsymbol{v}_2$ with $\boldsymbol{v}_1 \in \mathcal{V}$, $\boldsymbol{v}_2 \in \mathcal{V}^\top$ and $\|\boldsymbol{v}_1\|, \|\boldsymbol{v}_2\| \leq 1$. Now, if $\boldsymbol{v}_1 = 0$, then $\boldsymbol{N}\boldsymbol{v} = \boldsymbol{v}$, and hence $\boldsymbol{v}$ is an eigenvector with eigenvalue 1. Instead, if $\boldsymbol{v}_1 \neq 0$, then there exists $i' \in \{1, \dots, k\}$ (we take the largest one) such that $\beta_{i'} \in (0,2)$ and $(\boldsymbol{v}^\top \boldsymbol{z}_{i'})^2 = (\boldsymbol{v}_1^\top \boldsymbol{z}_{i'})^2 \in (0,1]$. Therefore, if $i' < k$, then either $\beta_j = 0$ or $\boldsymbol{z}_j^\top \boldsymbol{v} = 0$ so that $\boldsymbol{C}_j \boldsymbol{v} = \boldsymbol{v}$ for all $j \in \{i'+1, \dots, k\}$. Moreover, we have that

$$\|\boldsymbol{C}_{i'}\boldsymbol{v}\|^2 = \|\boldsymbol{v} - \beta_{i'}\boldsymbol{z}_{i'}\boldsymbol{z}_{i'}^\top\boldsymbol{v}\|^2 = 1 - \beta_{i'}(2-\beta_{i'})(\boldsymbol{v}^\top \boldsymbol{z}_{i'})^2 < 1,$$

where the last line comes from the fact that $\min_{x\in[0,2]} x(2-x) = 0$ and is only reached at $x = 0$ and $x = 2$, while $\beta_{i'} \in (0,2)$. Therefore, since for every $i$, $\|\boldsymbol{C}_i\| \leq 1$ and the Euclidean norm is sub-multiplicative we have

$$\|\boldsymbol{N}\boldsymbol{v}\| = \|\boldsymbol{C}_1 \boldsymbol{C}_2 \dots \boldsymbol{C}_k \boldsymbol{v}\| = \|\boldsymbol{C}_1 \boldsymbol{C}_2 \dots \boldsymbol{C}_{i'}\boldsymbol{v}\| \leq \|\boldsymbol{C}_1\| \cdots \|\boldsymbol{C}_{i'}\boldsymbol{v}\| < 1.$$

Therefore, if $\boldsymbol{v}$ is also an eigenvector with eigenvalue $\lambda \in \mathbb{C}$, then $\|\boldsymbol{N}\boldsymbol{v}\| = |\lambda| < 1$. Hence, we proved that for every eigenvector with eigenvalue $\lambda$ either $\lambda = 1$ or $|\lambda| < 1$.

It remains to show that all eigenvalues of $\boldsymbol{N} \in \mathcal{M}_2^n([0,1])$ are in $[0,1]$. From the assumptions $\boldsymbol{N} = \boldsymbol{C}_1 \boldsymbol{C}_2$ with $\boldsymbol{C}_1, \boldsymbol{C}_2$ symmetric and positive semi-definite, therefore $\boldsymbol{C}_1$ has a unique symmetric and positive semi-definite square root $\boldsymbol{C}_1^{1/2}$ such that $\boldsymbol{C}_1^{1/2}\boldsymbol{C}_1^{1/2} = \boldsymbol{C}_1$. If $\boldsymbol{C}_1$ is non-singular (invertible) then

$$\boldsymbol{C}_1 \boldsymbol{C}_2 = \boldsymbol{C}_1^{1/2}\boldsymbol{C}_1^{1/2}\boldsymbol{C}_2\boldsymbol{C}_1^{1/2}\boldsymbol{C}_1^{-1/2}.$$

---

[4]This is a specialization of the Cartan–Dieudonné Theorem to $\mathbb{R}^n$, see Theorem 3 in https://faculty.uml.edu/dklain/orthogonal.pdf for a proof.

Thus, $C_1 C_2$ is similar to $C_1^{1/2} C_2 C_1^{1/2}$ and shares its eigenvalues. Moreover $C_1^{1/2} C_2 C_1^{1/2}$ is symmetric positive semi-definite (having real nonnegative eigenvalues) because $C_1^{1/2}$ and $C_2$ are symmetric and $\boldsymbol{v}^\top C_1^{1/2} C_2 C_1^{1/2} \boldsymbol{v} = \boldsymbol{z}^\top C_2 \boldsymbol{z} \geq 0$ with $\boldsymbol{z} = C_1^{1/2} \boldsymbol{v}$ since $C_2$ is positive semi-definite. Instead, if $C_1$ is singular, for $t > 0$ the matrix $C_1 + t\boldsymbol{I}$ is positive definite and non-singular. Hence $(C_1 + t\boldsymbol{I}) C_2$ has real and nonnegative eigenvalues. Since $C_1 C_2 = \lim_{t\to 0}(C_1 + t\boldsymbol{I})(C_2)$ and the eigenvalues are a continuous function of the entry of the matrix, $C_1 C_2$ has positive real eigenvalues. Since the modulus of any eigenvalue is smaller or equal than the euclidean norm of the matrix, which is smaller than one from the first point of the theorem, the statement follows. $\square$

## C.3 Proof of Theorem 3

We first recall the notion of group isomorphism. Two groups $(G, *)$ and $(H, \cdot)$ where $G, H$ are the sets and $\star$ and $\cdot$ are the associative operations, are isomorphic, if there exists a bijective map $f : G \to H$ such that for every $g \in G, h \in H$

$$f(g * h) = f(g) \cdot f(h).$$

We view the LRNN layer in (1) as the automaton $\mathcal{A}_{\text{lin}} = (\Sigma, \mathcal{H}, \boldsymbol{H}_0, \delta_{\text{lin}})$, where $\delta_{\text{lin}}(\boldsymbol{H}, w) = \boldsymbol{A}(w)\boldsymbol{H} + \boldsymbol{B}(w)$, which is extended in the usual way, and $\mathcal{H} = \{\delta_{\text{lin}}(\boldsymbol{H}_0, w) : w \in \Sigma^*\}$. Since we assumed that $\mathcal{T}(\mathcal{A})$ is a group, from Cayley's theorem we have that it is isomorphic to a subgroup of $S_n$, which is the set of permutations on a set of $n$ elements. Furthermore, each element in $S_n$ can be represented as an $n \times n$ permutation matrix. Since in general $n \neq |Q|$, we cannot let $\mathcal{H}$ to be a set of one hot vectors each corresponding to states in $Q$. Instead, we let $\boldsymbol{H}_0 = (1, \ldots, n)^\top$, $\mathcal{P} \subset \{0,1\}^{n\times n}$ be the set of permutation matrices and set $\boldsymbol{B} \equiv 0$ and $\boldsymbol{A} : \Sigma \to \mathcal{P}$ to be the function mapping each letter $w \in \Sigma$ to the permutation matrix corresponding to $\delta(\cdot, w)$. With this choice we can see that the function $f : \mathcal{T}(\mathcal{A}_{\text{lin}}) \to \mathcal{T}(\mathcal{A})$ such that $f(\delta_{\text{lin}}(\cdot, \boldsymbol{w})) = \delta(\cdot, \boldsymbol{w})$ for every $\boldsymbol{w} \in \Sigma^*$ is one-to-one (bijective), and from our choice of $\boldsymbol{H}_0$, the map $h : \mathcal{T}(\mathcal{A}_{\text{lin}}) \to \mathcal{H}$ such that for every $\boldsymbol{w} \in \Sigma^*$, $h(\delta_{\text{lin}}(\cdot, \boldsymbol{w})) = \delta_{\text{lin}}(\boldsymbol{H}_0, \boldsymbol{w})$ is also bijective. Moreover, the map $\phi : \mathcal{T}(\mathcal{A}) \to Q$ such that $\phi(\delta(\cdot, \boldsymbol{w})) = \delta(q_0, \boldsymbol{w})$ is surjective because without loss of generality we can consider states that are only reachable from the initial state $q_0$, i.e. $Q = \{\delta(q_0, \boldsymbol{w}) : \boldsymbol{w} \in \Sigma^*\}$. Hence if we set $g = \phi \circ f \circ h^{-1}$, then $g : \mathcal{H} \to Q$ is surjective and for every $w \in \Sigma$ and $\boldsymbol{H} \in \mathcal{H}$ we have that

$$g(\delta_{\text{lin}}(\boldsymbol{H}, w)) = \delta(g(\boldsymbol{H}), w)$$

Thus, we have shown that such an LRNN implements $\mathcal{A}$ and it does so with finite precision because the entries of all vectors and matrices are bounded integers. Moreover, Let $k = \max_{w\in\Sigma} \sum_{q\in Q} \mathbf{1}\{\delta(q, w) \neq q\} = \max_{w\in\Sigma} \sum_{i=1}^n \mathbf{1}\{(\boldsymbol{A}(w)\boldsymbol{H}_0)_i = \boldsymbol{H}_{0,i}\}$ be the maximum number of displaced element of the permutation associated with the alphabet $\Sigma$. Then, this means that each permutation can be written as a product of at most $k-1$ permutations of two elements. Hence, for every $w \in \Sigma$, $\boldsymbol{A}(w) \in \mathcal{M}_{k-1}^n(\{-1, 1\})$.

If in addition there exists $m \in \mathbb{N}$ such that $\mathcal{T}(\mathcal{A})$ is isomorphic to a subgroup of the cyclic group $\mathbb{Z}_m$ with elements $\{0, \ldots, m-1\}$, we can modify the construction above to use a smaller dimension. If $m = 2$, then $\mathbb{Z}_2$ has elements $\{0, 1\}$, and $\mathcal{A}$ implements the parity automaton. Thus, we can set $\boldsymbol{H}_0 = -1$, $\boldsymbol{A}(0) = 1$, $\boldsymbol{A}(1) = -1$ and $g(1) = 1$ while $g(-1) = 0$, which means that we can use a scalar recursion. Otherwise, if $m \geq 3$, we can modify the construction above by setting $\boldsymbol{H}_0 = (1, 0)^\top$ and, if for simplicity we assume $\Sigma \in \{0, \ldots, m-1\}$, for every $w \in \Sigma$ we let $\boldsymbol{A}(w)$ be the $2 \times 2$ rotation matrix corresponding to $\delta(\cdot, w)$:

$$\boldsymbol{A}(w) = \boldsymbol{R}(\theta_w) = \begin{bmatrix} \cos\theta_w & -\sin\theta_w \\ \sin\theta_w & \cos\theta_w \end{bmatrix}, \quad \theta_w = \frac{2\pi w}{m},$$

such that $\boldsymbol{R}(\theta_w) \in \mathcal{M}_2^2(\{-1\})$ (from Proposition 1). This concludes the proof. $\square$

## C.4 Krohn-Rhodes Theorem

Before presenting the proof for Theorem 4, we provide the statement for the landmark result of Krohn-Rhodes (Krohn & Rhodes, 1965), after giving the definition of the cascade product of two FSA.

**Definition 1** (Cascade product). *Given two FSA $\mathcal{A} = (\Sigma, Q, q_0, \delta)$ and $\mathcal{B} = (Q \times \Sigma, Q', q_0', \delta')$, we define the cascade product FSA as $\mathcal{C} = \mathcal{B} \circ \mathcal{A} = (\Sigma, Q \times Q', (q_0, q_0'), \delta'')$ where for any $w \in \Sigma$*

$$\delta''((q, q'), w) := (\delta(q, w), \delta'(q', (q, w)))$$

**Theorem 5** (Krohn-Rhodes, Theorem 4 in Maler & Pnueli (1994)). *For every FSA $\mathcal{A} = (\Sigma, Q, q_0, \delta)$ there exists $s \leq 2^{|Q|}$ and a cascade product FSA $\mathcal{C} = \mathcal{A}^{(s)} \circ \cdots \circ \mathcal{A}^{(1)} = (\Sigma, Q^\times, q_0^\times, \delta^\times)$, with $\mathcal{A}^{(i)} = (\Sigma^{(i)}, Q^{(i)}, q_0^{(i)}, \delta^{(i)})$, with $|Q^{(i)}| \leq |Q|$, and a function $\mathcal{W} : Q^\times \to Q$ such that for any $\boldsymbol{w} \in \Sigma^*$, $\delta(q_0, \boldsymbol{w}) = \mathcal{W}(\delta^\times(q_0^\times, \boldsymbol{w}))$ and each $\mathcal{A}^{(i)}$ is permutation-reset automaton, which means that for every $w^{(i)} \in \Sigma^{(i)}$, $\delta^{(i)}(\cdot, w^{(i)})$ is either a bijection (i.e. a permutation over $Q$) or constant, ie. $\delta(\cdot, w^{(i)}) = q(w^{(i)}) \in Q^{(i)}$.*

### C.5 Proof of Theorem 4

We apply the Krohn-Rhodes theorem (Theorem 5) to write $\mathcal{A}$ as the cascade product FSA $\mathcal{C} = \mathcal{A}^{(s)} \circ \cdots \circ \mathcal{A}^{(1)}$ with each FSA $\mathcal{A}^{(i)} = (\Sigma^{(i)}, Q^{(i)}, q_0^{(i)}, \delta^{(i)})$ being permutation-reset and we show how the LRNN can implement $\mathcal{C}$ by first showing how its $i$-th layer, with the structure in (1), can implement $\mathcal{A}^{(i)}$.

Let $n = |Q^{(i)}|$ and without loss of generality assume that $\Sigma = \{1, 2, \ldots, |\Sigma|\}$ and $Q^{(i)} = \{1, 2, \ldots, n\}$ with $q_0^{(i)} = 1$. For every $w \in \Sigma^{(i)}$ we set $\boldsymbol{A}^{(i)}(w) \in \{0,1\}^{n \times n}$, $\boldsymbol{B}^{(i)}(w) \in \{0,1\}^n$ such that for every $q, q' \in Q^{(i)}$

$$\boldsymbol{A}^{(i)}(w)_{q',q} = \mathbf{1}\{\delta(q, w) = q'\}, \quad \boldsymbol{B}^{(i)}(w)_{q'} = 0, \qquad \text{if } \delta^{(i)}(\cdot, w) \text{ is bijective, or}$$

$$\boldsymbol{A}^{(i)}(w)_{q',q} = 0, \qquad\qquad \boldsymbol{B}^{(i)}(w)_{q'} = \mathbf{1}\{q' = q(w)\}, \quad \text{if } \delta^{(i)}(\cdot, w) \equiv q(w).$$

Then, for every word $\boldsymbol{w}^{(i)} = w_1^{(i)} \ldots w_t^{(i)} \in \Sigma^{(i)*}$, we set $g : \mathbb{R}^n \to \mathbb{R}$, such that $g(x) = (1, \ldots, n)^\top x$ and

$$\boldsymbol{H}_t^{(i)} = \boldsymbol{A}^{(i)}(w_t^{(i)})\boldsymbol{H}_{t-1}^{(i)} + \boldsymbol{B}^{(i)}(w_t^{(i)}), \qquad \boldsymbol{H}_0^{(i)} = (1, 0 \ldots, 0)^\top \in \mathbb{R}^n$$

$$y^{(i)} = \mathrm{dec}^{(i)}(\boldsymbol{H}_t^{(i)}, w_t^{(i)}) = (g(\boldsymbol{H}_t^{(i)}), w_t^{(i)}) = (\delta^{(i)}(q_0^{(i)}, \boldsymbol{w}^{(i)}), w^{(i)})$$

So that such construction implements $\mathcal{A}^{(i)}$. In addition, by letting $\boldsymbol{w} = w_1 \ldots w_t \in \Sigma^*$ be the input to the LRNN, i.e. $w_j^{(1)} = w_j$, and setting the output of each layer as the input to the next, i.e. $w_j^{(i)} = y_j^{(i-1)}$ for $i \geq 2$, for the output of the last layer we get

$$\begin{aligned}
y_t^{(s)} &= \mathrm{dec}^{(s)}(\boldsymbol{H}_t, w_t^{(s)}) \\
&= (\delta^{(s)}(q_0^{(s)}, \boldsymbol{w}^{(s)}), y_t^{(s-1)}) \\
&= (\delta^{(s)}(q_0^{(s)}, \boldsymbol{w}^{(s)}), \delta^{(s-1)}(q_0^{(s-1)}, \boldsymbol{w}^{(s-1)}), y_t^{(s-2)}) \\
&= (\delta^{(s)}(q_0^{(s)}, \boldsymbol{w}^{(s)}), \ldots, \delta^{(1)}(q_0^{(1)}, \boldsymbol{w}), w_t) \in \mathbb{N}^{s+1},
\end{aligned}$$

where we removed the nested parenthesis for simplicity. Hence, the first $s$ elements of $y_t^{(s)}$ are exactly the output of the cascade FSA $\mathcal{C}$. Note that our construction can be implemented in finite precision since we only used matrices/vectors with entries either in $\{0, 1\}$, requiring only one bit, or in $Q^{(i)} \subset \mathbb{N}$, that can also be implemented using finite precision with $|Q^{(i)}|$ integers, requiring $\log_2(|Q^{(i)}|)$ bits. Also note that we can exclude $w_t$ from the output $y_t^{(s)}$ by changing $\mathrm{dec}^{(s)}$, to bring the dimension of the output, end hence the width of the LRNN, to $\mathbb{N}^s$.

It is also the case that $\|\boldsymbol{A}^{(i)}(w)\| \leq 1$ for every $w \in \Sigma^{(i)}$ since $\boldsymbol{A}^{(i)}(w)$ is either a permutation matrix ($\|\boldsymbol{A}^{(i)}(w)\| = 1$) or the zero matrix ($\|\boldsymbol{A}^{(i)}(w)\| = 0$). Also, for every permutation matrix $\boldsymbol{P} \in \{0,1\}^{n \times n}$ which permutes only $k \leq n$ elements we have that $\boldsymbol{P} \in \mathcal{M}_{k-1}^n(\{-1, 1\})$.

Furthermore, for the zero matrix, we have

$$0 = \prod_{i=1}^{n}(I - \boldsymbol{e}_i \boldsymbol{e}_i^\top) \in \mathcal{M}_n^n(\{0\})$$

It follows that $\mathcal{A}^{(i)}(w) \in \mathcal{M}_n^n([-1, 1])$ for every $i \in \{1, \ldots, s\}$ and $w \in \Sigma^{(i)}$. $\qquad\square$

## D LRNNs CAN DO MODULAR ADDITION USING ONLY REFLECTIONS

In this section, we explain how an LRNN with two layers and using only Householder state transition matrices (reflections) can compute addition modulo $m \in \mathbb{N}$, i.e it can map words $x_1, \ldots, x_t$ with $x_i \in \{0, \ldots, m-1\}$ into $y_t = (\sum_{i=1}^m x_i) \bmod m$ for arbitrary $t \in \mathbb{N}$. This corresponds to solving the group word problem associated with the cyclic group $\mathbb{Z}_m$. We note that our modification of DeltaNet, namely DeltaNet $[-1, 1]$ can therefore solve addition modulo $m$ with 2 layers.

If the state transition matrices can be generic rotation matrices, then an LRNN can perform addition modulo $m$ using just one layer by mapping each element of $\mathbb{Z}_m$ to the corresponding $2 \times 2$ rotation matrix as shown in Appendix C.3. Such construction requires a number of states for the LRNN equal to $m$, i.e. the number of elements of the group $\mathbb{Z}_m$. However, since here we assume that state transition matrices are reflections, we cannot map each element of the group to a rotation (since those are a product of 2 reflections) and our construction for the LRNN will require two layers. Specifically, the first layer will count modulo 2, i.e. it will output the sequence $\boldsymbol{y}_1^{(1)}, \ldots, \boldsymbol{y}_t^{(1)}$ where $\boldsymbol{y}_i^{(1)} = (x_i, i \bmod 2)$, while the second layer will have $2m$ states and will use two different reflection matrices for each group element, depending on the value of $y_{i,2}^{(1)} = i \bmod 2$. Formally, we have the following result.

**Theorem 6** (Modular addition with reflections). *An LRNN with two layers in the form (1), where $\boldsymbol{A} : \mathbb{N} \to \{-1\}$ for the first layer and $\boldsymbol{A} : \mathbb{R}^2 \to \mathcal{M}_1^2(\{-1\})$ for the second layer, with $\mathcal{M}_1^2$ defined in (3), can perform addition modulo $m$ for any $m \in \mathbb{N}$. In particular, the LRNN will have 2 scalar states in the first layer and $2m$ states, each being a vector in $\mathbb{R}^2$, in the second layer.*

*Proof.* The first layer of the LRNN will implement counting modulo 2 as follows.
$$h_0^{(1)} = 0, \quad h_t^{(1)} = -h_{t-1}^{(1)} + 1, \quad \boldsymbol{y}_t^{(1)} = \text{dec}^{(1)}(h_t, x_t) = (x_t, h_t).$$
We note that the state-transition matrix (the scalar $-1$) is a reflection since $\{-1\} = \mathcal{M}_1^1(\{-1\})$. For the second layer, we have instead
$$\boldsymbol{h}_0^{(2)} = (1, 0)^\top, \quad \boldsymbol{h}_t^{(2)} = \boldsymbol{A}^{(2)}(\boldsymbol{y}_t^{(1)})\boldsymbol{h}_{t-1}^{(2)}, \quad \boldsymbol{y}_t^{(2)} = \text{dec}^{(2)}(\boldsymbol{h}_t^{(2)}, \boldsymbol{y}_t^{(1)})$$
$$\boldsymbol{A}^{(2)}(\boldsymbol{y}) = \boldsymbol{H}(\theta(y_1, y_2)) = \begin{bmatrix} \cos\theta(y_1, y_2) & \sin\theta(y_1, y_2) \\ \sin\theta(y_1, y_2) & -\cos\theta(y_1, y_2) \end{bmatrix}$$
$$\text{dec}^{(2)}(\boldsymbol{h}, \boldsymbol{y}) = \underset{i \in \{0, \ldots, m-1\}}{\arg\max} \max(\boldsymbol{c}_i^\top \boldsymbol{h}, \boldsymbol{d}_i^\top \boldsymbol{h})$$
where $\boldsymbol{y} = (y_1, y_2)^\top \in \{0, \ldots, m-1\} \times \{0, 1\}$, $\boldsymbol{H}(\alpha)$ is the $2 \times 2$ reflection matrix that reflects all vectors by a line having an angle of $\alpha/2$ with the line passing from the origin and the vector $(1, 0)^\top$ and $\theta : \{0, \ldots, m-1\} \times \{0, 1\} \to \mathbb{R}$ determines the angle of the reflection and is defined as
$$\theta(i, 1) = \frac{(1 - 2i)\pi}{m}, \quad \theta(i, 0) = \frac{(1 + 2i)\pi}{m}, \quad \text{for all } i \in \{0, \ldots, m-1\}.$$
Moreover $\mathcal{C} = \{\boldsymbol{c}_0, \ldots, \boldsymbol{c}_{m-1}\}$ and $\mathcal{D} = \{\boldsymbol{d}_0, \ldots, \boldsymbol{d}_{m-1}\}$ are the two sets of states corresponding to reflections and rotations respectively and are defined as
$$\boldsymbol{d}_0 = \boldsymbol{h}_0^{(2)} = (1, 0)^\top, \quad \boldsymbol{c}_0 = \boldsymbol{H}(\pi/m)\boldsymbol{d}_0,$$
$$\boldsymbol{d}_i = \boldsymbol{R}(2i\pi/m)\boldsymbol{d}_0, \quad \boldsymbol{c}_i = \boldsymbol{R}(-2i\pi/m)\boldsymbol{c}_0 \quad \text{for all } i \in \{0, \ldots, m-1\},$$
where $\boldsymbol{R}(\beta)$ is a rotation matrix with angle $\beta \in \mathbb{R}$.

Let $\alpha, \gamma \in \mathbb{R}$, the following are standard identities of products of 2D rotations and reflections.
$$\boldsymbol{R}(\alpha)\boldsymbol{R}(\gamma) = \boldsymbol{R}(\alpha + \gamma), \quad \boldsymbol{H}(\alpha)\boldsymbol{H}(\gamma) = \boldsymbol{R}(\alpha - \gamma),$$
$$\boldsymbol{R}(\alpha)\boldsymbol{H}(\gamma) = \boldsymbol{H}(\alpha + \gamma) \quad \boldsymbol{H}(\gamma)\boldsymbol{R}(\alpha) = \boldsymbol{H}(\gamma - \alpha).$$
From our choice of $\theta$, $\boldsymbol{d}_i$ and $\boldsymbol{c}_i$, using the identities above and the the fact that $\boldsymbol{R}$ is a periodic function with period $2\pi$ we have that
$$\begin{aligned} \boldsymbol{H}(\theta(j, 1))\boldsymbol{d}_i &= \boldsymbol{H}(\theta(j, 1))\boldsymbol{R}(2i\pi/m)\boldsymbol{d}_0 \\ &= \boldsymbol{H}(\theta(j, 1))\boldsymbol{R}(2i\pi/m)\boldsymbol{H}(\pi/m)\boldsymbol{c}_0 \\ &= \boldsymbol{H}(\theta(j, 1))\boldsymbol{H}(\theta(i, 0))\boldsymbol{c}_0 \\ &= \boldsymbol{R}(\theta(j, 1) - \theta(i, 0))\boldsymbol{c}_0 \\ &= \boldsymbol{R}(-2(i + j)\pi/m)\boldsymbol{c}_0 = \boldsymbol{c}_{i+j \bmod m}, \end{aligned} \quad (6)$$

and similarly

$$\begin{aligned}
\boldsymbol{H}(\theta(j,0))\boldsymbol{c}_i &= \boldsymbol{H}(\theta(j,1))\boldsymbol{R}(-2i\pi/m)\boldsymbol{c}_0 \\
&= \boldsymbol{H}(\theta(j,0))\boldsymbol{R}(-2i\pi/m)\boldsymbol{H}(\pi/m)\boldsymbol{d}_0 \\
&= \boldsymbol{H}(\theta(j,0))\boldsymbol{H}(\theta(i,1))\boldsymbol{d}_0 \\
&= \boldsymbol{R}(\theta(j,0)-\theta(i,1))\boldsymbol{d}_0 \\
&= \boldsymbol{R}(2(i+j)\pi/m)\boldsymbol{d}_0 = \boldsymbol{d}_{i+j \bmod m},
\end{aligned} \tag{7}$$

for every $i,j \in \{0,\dots,m-1\}$. We will now prove by induction that

$$\boldsymbol{h}_t^{(2)} = \begin{cases} \boldsymbol{c}_{y_t} & \text{if } t \bmod 2 = 1 \\ \boldsymbol{d}_{y_t} & \text{if } t \bmod 2 = 0 \end{cases}. \tag{8}$$

where we recall that $y_i := (\sum_{j=1}^i x_j) \bmod m$ and that, by definition, $\boldsymbol{h}_0^{(2)} = \boldsymbol{d}_0$ and $\boldsymbol{h}_i^{(2)} = \boldsymbol{H}(\theta(x_i, i \bmod 2))\boldsymbol{h}_{i-1}^{(2)}$, since $\boldsymbol{y}_i^{(1)} = (x_i, i \bmod 2)$. For the base case we have that

$$\boldsymbol{h}_1^{(2)} = \boldsymbol{H}(\theta(x_1,1))\boldsymbol{h}_0^{(2)} = \boldsymbol{H}(\theta(x_1,1))\boldsymbol{d}_0 = \boldsymbol{c}_{x_1 \bmod m} = \boldsymbol{c}_{y_1}$$
$$\boldsymbol{h}_2^{(2)} = \boldsymbol{H}(\theta(x_2,0))\boldsymbol{h}_1^{(2)} = \boldsymbol{H}(\theta(x_2,0))\boldsymbol{c}_{x_1 \bmod m} = \boldsymbol{d}_{x_1+x_2 \bmod m} = \boldsymbol{d}_{y_2},$$

where we have used (6) and (7). As induction hypothesis, suppose that for $i \geq 2$

$$\boldsymbol{h}_i^{(2)} = \begin{cases} \boldsymbol{c}_{y_i} & \text{if } i \bmod 2 = 1 \\ \boldsymbol{d}_{y_i} & \text{if } i \bmod 2 = 0 \end{cases}$$

then, using again (6) and (7), we obtain

$$\boldsymbol{h}_{i+1}^{(2)} = \begin{cases} \boldsymbol{H}(\theta(x_{i+1},1))\boldsymbol{h}_i^{(2)} = \boldsymbol{H}(\theta(x_{i+1},1))\boldsymbol{c}_{y_i} = \boldsymbol{c}_{x_{i+1}+y_i \bmod m} = \boldsymbol{c}_{y_{i+1}} & \text{if } i \bmod 2 = 1 \\ \boldsymbol{H}(\theta(x_{i+1},0))\boldsymbol{h}_i^{(2)} = \boldsymbol{H}(\theta(x_{i+1},0))\boldsymbol{d}_{s_i} = \boldsymbol{d}_{x_{i+1}+y_i \bmod m} = \boldsymbol{d}_{y_{i+1}} & \text{if } i \bmod 2 = 0 \end{cases}.$$

which completes our proof by induction yielding (8). Finally, using the definition of $\mathrm{dec}^{(2)}$, (8) and as long as $\boldsymbol{d}_i \neq \boldsymbol{c}_j$, $\boldsymbol{d}_i \neq \boldsymbol{d}_j$ and $\boldsymbol{c}_i \neq \boldsymbol{c}_j$ for every $i,j$ with $i \neq j$, which is guaranteed by our choice of $\theta$, we have that $\mathrm{dec}^{(2)}(\boldsymbol{h}_t^{(2)}, \boldsymbol{y}_t^{(1)}) = (\sum_{j=1}^i x_j) \bmod m = y_t$, ending the proof. □

# E EXPERIMENTS

## E.1 CHOMSKY HIERARCHY

Here, we provide details on the formal language tasks and experimental protocol of Section 5.1.

### E.1.1 DETAILS ON THE EXPERIMENTAL SETUP

Like Beck et al. (2024), we trained each model with sequence lengths ranging from 3 to 40 and evaluated on lengths from 40 to 256, to understand the length generalization capabilities. We compared mLSTM and sLSTM with two models: Mamba (Gu & Dao, 2024) and DeltaNet (Yang et al., 2024b). Moreover, we also include a Transformer (Vaswani et al., 2017) baseline. For parity, all models contain 2 blocks (layers), with 4 heads for the xLSTM and DeltaNet models. We set the embedding and heads' dimensions to 128. For Mamba and DeltaNet, we also enable the 1-D depthwise-separable convolution layer with kernel size equal to 4 after the query/key/value projection. For modular arithmetic, we increase the number of layers to 3 and use a gradient clipping norm of 1.0 for Transformer, Mamba, and DeltaNet, while for mLSTM and sLSTM we decrease the embedding size and number of heads to 64 and 1, respectively, as well as use a standard initialization for the bias parameters. We train each model using AdamW (Loshchilov & Hutter, 2019) without gradient clipping, using 3 different learning rates (1e-2, 1e-3, 5e-4 1e-4), with 3 different seeds each. We pick the best based on the median of the 3 seeds for every learning rate value. We use a batch size of 1024 (except for mLSTM, where we use 512 due to OOM error) and a cosine annealing learning rate schedule (Loshchilov & Hutter, 2017) (minimum learning rate: 1e-6) after 10% warm-up steps. The weight decay is set to 0.1 during training. We train on every task for 100k steps in total. At each training step, we make sure to generate a valid random sample from the task at hand (see below).

Table 5: Performance comparison of various recurrent models on regular and context-free language tasks. recurrent models on formal language tasks. We report the median $\pm$ median absolute deviation (*left table*) and best score (*right table*) of 3 independent runs with different random seeds. Scores represent scaled accuracy, with 1.0 indicating perfect performance and 0.0 random guessing. The positive impact of allowing negative eigenvalues ($[-1, 1]$ range) versus restricting to positive eigenvalues ($[0, 1]$ range) is evident across different model architectures.

| | Parity | Mod. Arithmetic (w/o brackets) | Mod. Arithmetic (w/ brackets) | Mod. Arithm. (w/ brackets, no mult) |
|---|---|---|---|---|
| Transformer | $0.003 \pm_{0.013}$ | $0.018 \pm_{0.009}$ | $0.064 \pm_{0.003}$ | $0.025 \pm_{0.000}$ |
| mLSTM | $0.018 \pm_{0.035}$ | $0.027 \pm_{0.013}$ | $0.114 \pm_{0.000}$ | $0.034 \pm_{0.001}$ |
| sLSTM | $1.000 \pm_{0.000}$ | $0.124 \pm_{0.000}$ | $0.163 \pm_{0.015}$ | $0.153 \pm_{0.020}$ |
| Mamba $[0, 1]$ | $0.000 \pm_{0.000}$ | $0.066 \pm_{0.029}$ | $0.116 \pm_{0.007}$ | $0.072 \pm_{0.008}$ |
| Mamba $[-1, 1]$ | $1.000 \pm_{0.000}$ | $0.214 \pm_{0.027}$ | $0.098 \pm_{0.009}$ | $0.126 \pm_{0.010}$ |
| DeltaNet $[0, 1]$ | $0.010 \pm_{0.005}$ | $0.214 \pm_{0.056}$ | $0.162 \pm_{0.018}$ | $0.113 \pm_{0.009}$ |
| DeltaNet $[-1, 1]$ | $0.999 \pm_{0.006}$ | $0.826 \pm_{0.146}$ | $0.227 \pm_{0.011}$ | $0.129 \pm_{0.016}$ |

### E.1.2 DETAILS ON THE EVALUATED TASKS

In Section 5.1 we conducted empirical evaluations on 3 tasks –parity, modular arithmetic without brackets and with brackets – from various levels of the Chomsky Hierarchy, as proposed by Deletang et al. (2023) and similarly used in xLSTM (Beck et al., 2024). Details for each task are given below, where $|\Sigma|$ is the vocabulary size and $Acc_{rand}$ is the accuracy of random guessing:

- **Parity** ($|\Sigma| = 2$, $Acc_{rand} = 0.5$). The parity $y_t \in \{0, 1\}$ of a sequence of ones and zeros $\boldsymbol{x} = x_1 \ldots x_t \in \{0, 1\}^t$ is equal to 1 (resp. 0) if the total number of ones in the sequence is odd (resp. even). It is equivalent to addition modulo 2, it can be computed by summing all previous values and then using the modulo 2 function as $y_t = (\sum_{i=1}^t x_i) \bmod 2$.

- **Modular Arithmetic w/o Brackets** ($|\Sigma| = 10$, $Acc_{rand} = 1/5$). Given a set of special tokens $\Sigma_s = \{+, -, *, =, [\text{PAD}]\}$ and a modulus $m \geq 1$, we set $\Sigma = \Sigma_s \cup \{0, \ldots, m-1\}$ and $y_t$ is equal to the result of the operations modulo $m$ in the sequence $\boldsymbol{x} = \boldsymbol{x}_1, \ldots, \boldsymbol{x}_t$ with $x_i \in \Sigma$. In our experiments $m = 5$. An example sequence is as follows:

$$2 - 3 - 3 * 2 = 3 \, [\text{PAD}]$$

- **Modular Arithmetic w/ Brackets,** ($|\Sigma| = 12$, $Acc_{rand} = 1/5$). Same definition as the modular arithmetic without brackets with a set of special tokens $\Sigma_s = \{+, -, *, =, ), (, [\text{PAD}]\}$. In our experiments $m = 5$. An example sequence is as follows:

$$((((3 + 3) + -1) + -2) - ((3 - (-3)) + ((1) + 4))) = 2 \, [\text{PAD}]$$

### E.2 STATE-TRACKING

### E.2.1 DETAILS OF THE EXPERIMENTS

For the experiments in Section 5.2, we map each element of the group $S_5$ to an integer from 0 to 119, where 0 corresponds to the identity permutation, and then construct inputs and output sequences of integers $x_1, \ldots x_t$ and $y_1, \ldots, y_t$ as follows

- **$S_5$** We sample $x_i$ uniformly at random from $\{0, \ldots, 119\}$. $y_i$ is computed as the product of the permutations corresponding to $x_1, \ldots, x_i$ applied in order from 1 to $i$.

- **$S_5$ only swaps** As $S_5$ but $x_i$ is sampled from the permutations that permute up to two elements (swaps and identity).

- **$S_5$ swaps, 3-permutations** As $S_5$ but $x_i$ is sampled from the permutations that permute up to three elements.

- **$S_5$ 4 tokens per transition** If $i \bmod 4 = 0$, then $x_i$ is sampled uniformly at random from $\{0, \ldots, 119\}$, otherwise $x_i = 120$ (special token). For $i > 3$, $y_{i+3}$ is the product of the permutations corresponding to $x_1, \ldots, x_i$, where 120 is treated as the identity permutation. $y_i = 0$ for $i \in \{1, 2, 3\}$.

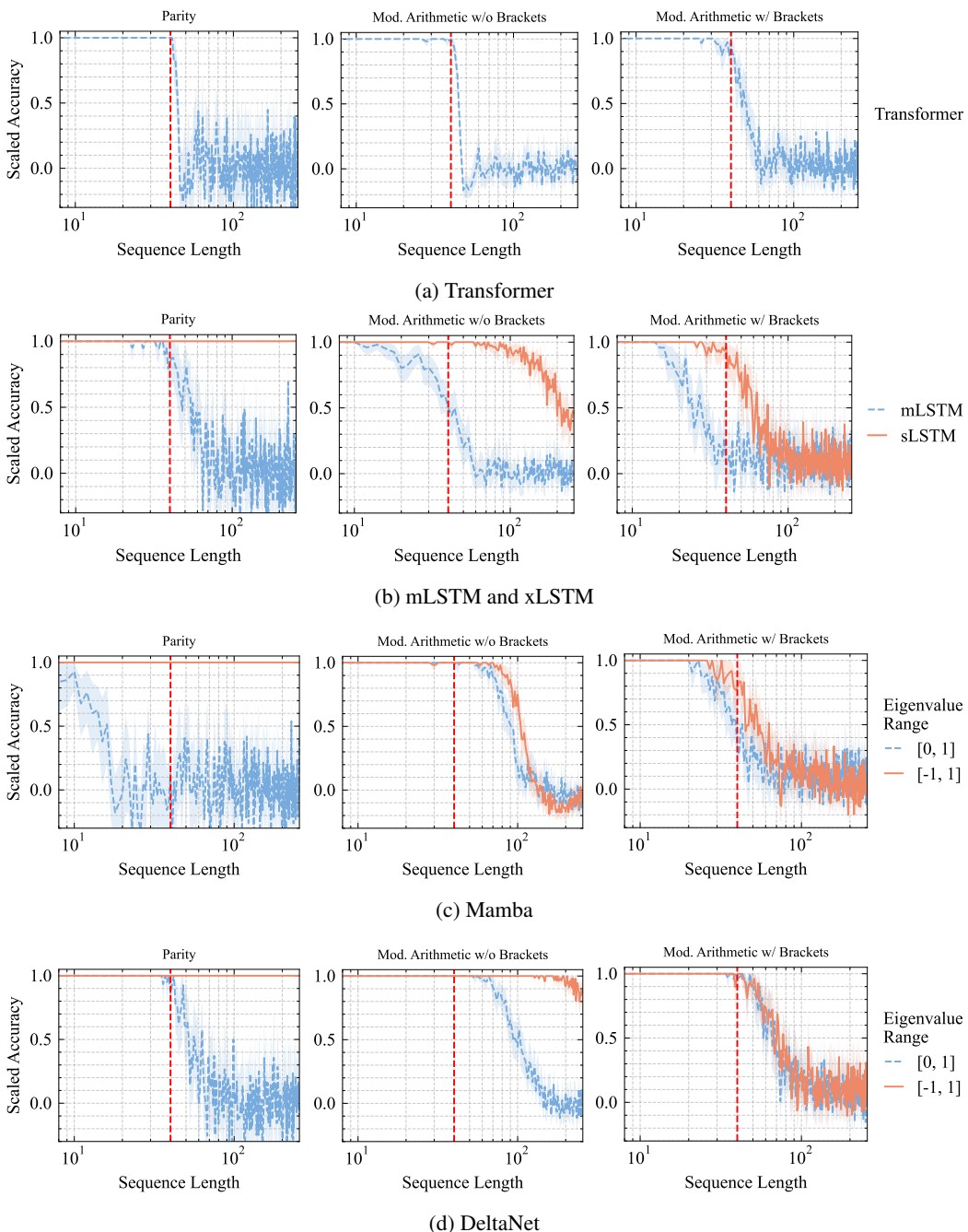

Figure 6: Performance (scaled accuracy) vs sequence length of *Transformer*, *mLSTM*, *sLSTM*, *Mamba* and *DeltaNet* variants on different formal language tasks. Trained on sequences up to length 40 (dashed vertical red line). At test time, we sample uniformly at random 8192 sequences with lengths between 40 and 256. The curves show the mean and 95% CI. Note, that the Transformer model fails to length extrapolate, but performs nearly perfectly within the training context length.

For each setup, we randomly sample $1.6$M examples for and $40K$ examples of length 500 to construct the train and test dataset. We note that we are using a substantially larger training set compared to (Merrill & Sabharwal, 2023), to reduce the chances of overfitting. We run 3 seeds for each method, changing the network initialization and sampling of the minibatches. The train and validation datasets are kept the same across runs.

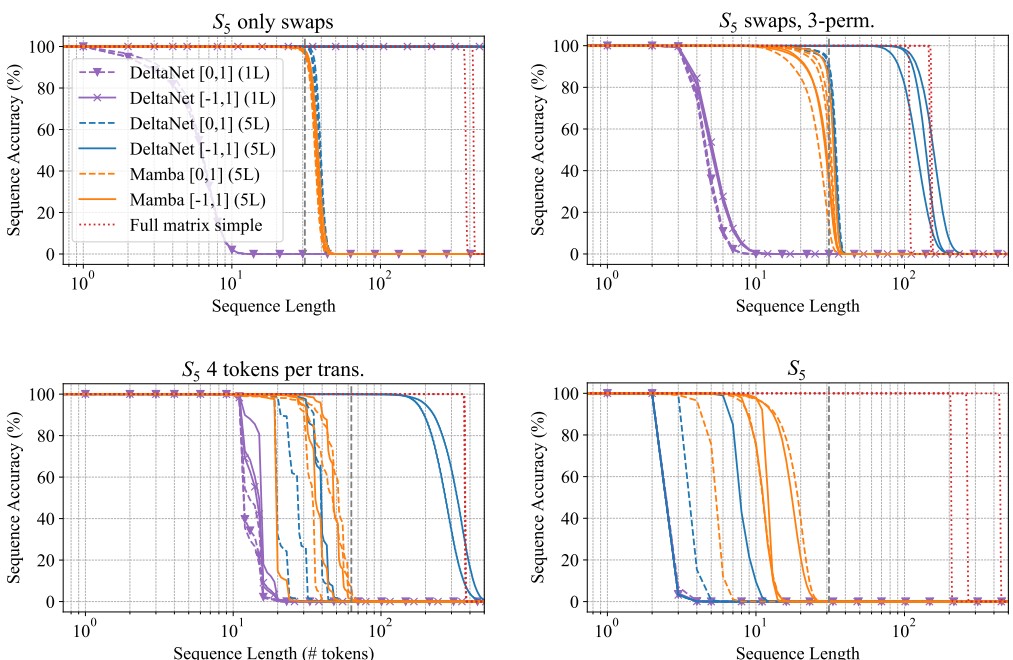

Figure 7: Validation sequence accuracy across different lengths on $S_5$ after 100 epochs of training (3 seeds). The dashed vertical line indicates the sequence length used during training. Each method is labeled with name, eigenvalue range, and number of layers. The dashed vertical line indicates the sequence length used during training.

We train all models using AdamW with weight decay $0.01$, learning rate $0.0001$, gradient clipping to $1.0$, and a batch size of $512$. Both DeltaNet and Mamba models use an embedding dimension of $128$ and $4$ heads for DeltaNet. In the case of DeltaNet, we do not use 1-D convolutions for these experiments. Other parameters are kept as default.

**Full Matrix Baseline.** For the full matrix baseline we use a single layer and map directly each token $x_i$ to a learnable full state-transition matrix $\boldsymbol{A}(x_i) \in \mathbb{R}^{n \times n}$ via one-hot encoding. We then compute, for $i \in \{1, \ldots, t\}$ the recursion

$$\boldsymbol{H}_i = \boldsymbol{A}(x_i)\boldsymbol{H}_{i-1}, \quad \boldsymbol{H}_0 = \boldsymbol{I} \in \mathbb{R}^{n \times n}$$

where $n$ is set to 32 for efficiency reasons (memory and compute time grow quickly with $n$). After that, we flatten each $\boldsymbol{H}_i$ into a vector and apply first a projection on the unit ball and then a linear decoder to get the final outputs. The projection was added to increase stability since we do not bound the norm of $\boldsymbol{A}(x_i)$. Since this model uses a full matrix, with $n \geq 5$ it should be fully able to learn $S_5$ without restricting the transitions in input or using more tokens per transition. However, in some situations, the performance degrades quickly after some input sequence length, probably because the norm of the learned $\boldsymbol{A}(x_i)$ is not close enough to one and hence part of the state either vanish or explode for long sequences.

**Plots with all runs.** We report the plots with all 3 runs per method in Figure 7 (In Figure 4 we reported only the best one for each method). Despite our efforts to decrease the variance of the results by increasing training time and dataset size, we report that there is still some variability. For example, one of the runs of DeltaNet $[-1, 1]$ (5L) on $S_5$ with 4 tokens per transition did not achieve a good accuracy.

### E.2.2 CYCLIC GROUPS

We report in Figure 8 some experiments on group word problems with the group $\mathbb{Z}_{60}$. For this experiment, we also consider the simplified version where each transition is encoded using 2 tokens. This is done as in the experiments of $S_5$ with 4 tokens, but using 2 tokens instead of 4. Extending the

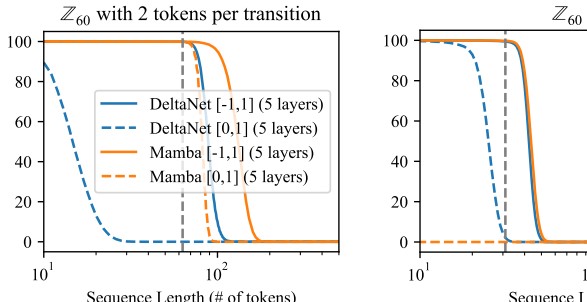

Figure 8: Validation sequence accuracy at different sequence lengths on the cyclic group $\mathbb{Z}_{60}$ (1 seed). Dashed vertical lines indicate the sequence length used for training (left 32, right 64). Using 2 tokens per transition seems to help only marginally in this case. Mamba $[-1, 1]$ is the best-performing model. The variants with eigenvalues in [0,1] performed worse.

eigenvalue range seems to help in both settings, although surprisingly, Mamba $[-1, 1]$, even though it has a diagonal state-transition matrix, seems to perform best. We conjecture that in this case, the models might learn the shortcut solutions, also because they do not generalize very well to longer sequences.

### E.3 Language Modeling

#### E.3.1 Details on the experimental setup

We use the training pipeline which is part of the flash-linear-attention library (flame) (Yang & Zhang, 2024) and which in turn is based on HuggingFace accelerate (Gugger et al., 2022). We use stage-2 of the ZeRO optimizer (Rajbhandari et al., 2020) with gradient clipping set to auto. The 1.3B parameter DeltaNet models are trained on 32 Nvidia A100s using a per-device batch size of 6 and 5 gradient accumulation steps for 50,000 steps. The 340M parameter DeltaNet models and the 370M parameter Mamba models are trained using a training batch size of 16 and 200,000 steps on 16 Nvidia A100s. All models are trained using a context length of 2048, learning rate of 3e-4. For optimization, we use AdamW (Loshchilov & Hutter, 2019), the learning rate was adjusted using cosine annealing (Loshchilov & Hutter, 2017) following a linear warm-up period of 250/500 steps for the 340/370M and 1.3B parameter models respectively. We applied a weight decay of 0.01 throughout the training process.

#### E.3.2 Details on the evaluated tasks

To produce the results in Table 4, we use the lm-harness benchmark (Gao et al., 2024), focusing on the same tasks as Yang et al. (2024b): LAMBADA (LMB) (Paperno et al., 2016), PIQA (Bisk et al., 2020), HellaSwag (Hella.) (Zellers et al., 2019), Winogrande (Wino.) (Sakaguchi et al., 2021), and ARC-easy (ARC-e) and ARC-challenge (ARC-c) (Clark et al., 2018). Additionally, we evaluate the performance on recall-intensive tasks (like Yang et al. (2024b)), including FDA (Arora et al., 2023), SWDE (Lockard et al., 2019), and SQUAD (Rajpurkar et al., 2018), to provide a comprehensive evaluation of our models' capabilities.

### E.4 Implementation

We build on the original code for Mamba[5] and DeltaNet[6]. For DeltaNet, implementing the extended eigenvalue range is straightforward, since there is no need to modify the Triton kernel. However, Mamba requires modifications to the CUDA code of the associative scan for both forward and backward passes which however had no impact on computational cost. We ensured the accuracy of the modifications by comparing the results with a naive implementation using a for-loop. For initial testing of the extended eigenvalue range, we used the pure PyTorch implementation of Mamba by Torres (2024). We provide listings of the necessary code changes in Mamba and DeltaNet in Appendix E.4.1. For DeltaNet, this changes also $\boldsymbol{B}(x_t)$ in Table 1, multiplying it by 2.

---

[5] https://github.com/state-spaces/mamba
[6] https://github.com/sustcsonglin/flash-linear-attention

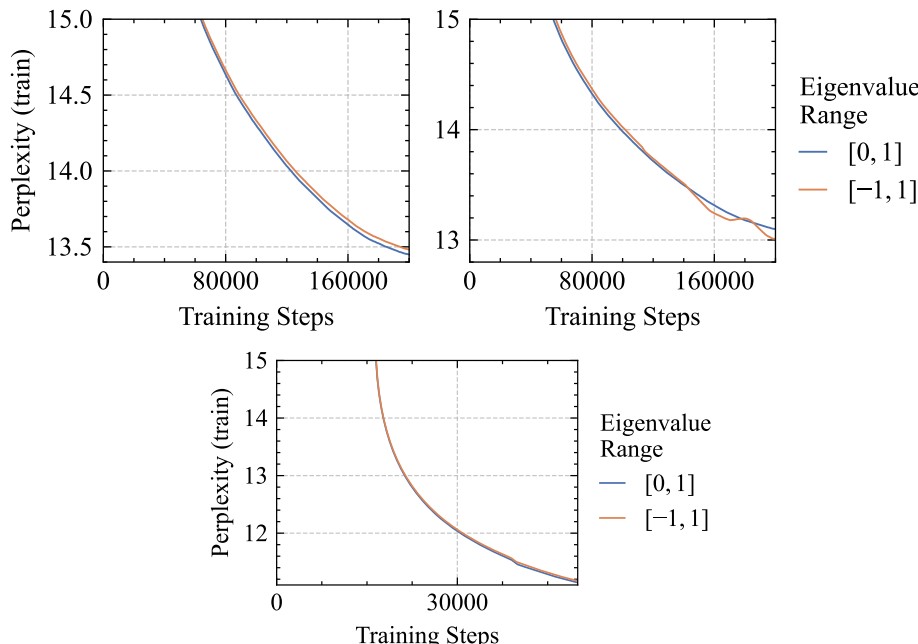

Figure 9: Learning curves of DeltaNet 340M (top left), Mamba 370M (top right) and DeltaNet 1.3B (bottom), training on 100B tokens of Fine-Web 100B. 1.3B runs required only 50k optimizer steps versus the 200k of the 340M runs due to the 4x larger batch size. All models trained stably with the same hyperparameters. Training curves were smoothed with a rolling window of 500 steps.

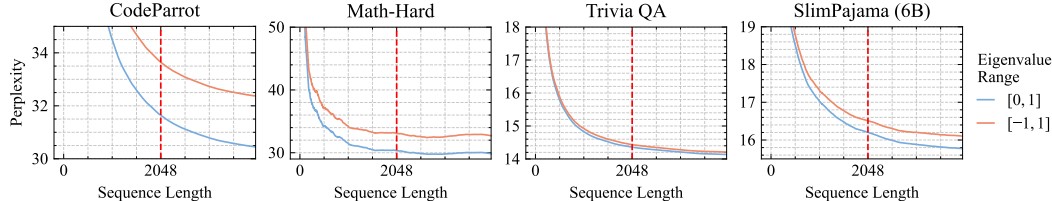

Figure 10: Length extrapolation performance of Mamba variants on different datasets. Mamba with eigenvalue range $[-1, 1]$ shows worse perplexity on coding and math tasks compared to the $[0, 1]$ baseline. The dashed, vertical line indicates the training context length of 2048 tokens.

**Products in Log-space** We note that some diagonal models such as Mamba2 (Dao & Gu, 2024), GLA (Yang et al., 2024a), and mLSTM (Beck et al., 2024) take advantage of the fact that all values of the state-transition matrices are positive to compute their repeated products in log-space. Our change would not allow us to do this directly, and early tests on the chunkwise parallel form of GLA showed degraded performance. Therefore, for this work, we decided to focus on Mamba and DeltaNet since they do not compute the products in log space. We mention however, that at the cost of increased computation time, it would be possible to do products in log space by converting each value in the diagonal state-transition matrix to the product of its absolute value and sign. This way, absolute values can be multiplied in log space, while products of signs are coincidentally equivalent to addition modulo 2, i.e. parity, and hence can be done stably. We leave the investigation of this approach to future work. Furthermore, we also believe that our change may be less suited for methods that use a normalized RNN state, such as mLSTM, since it might happen that the normalization term can be very close to zero due to the negative values.

### E.4.1 Implementation of Extended Eigenvalue Range

```
220  if constexpr (!kIsComplex) {
221  -   thread_data[i] = make_float2(exp2f(delta_vals[r][i] * A_val[r]),
222  +   thread_data[i] = make_float2(2.0f * exp2f(delta_vals[r][i] * A_val[r]) - 1.0f,
223                          !kIsVariableB ? delta_u_vals[r][i] : B_vals[i] * delta_u_vals[r][i]);
224      if constexpr (!Ktraits::kIsEvenLen) {
225          if (threadIdx.x * kNItems + i >= params.seqlen - chunk * kChunkSize) {
226              thread_data[i] = make_float2(1.f, 0.f);
227          }
228      }
229  }
```

Figure 11: Modifications to the forward pass of the Mamba associative scan. These changes extend the eigenvalue range from $[0, 1]$ to $[-1, 1]$, enhancing the model's expressive capacity. Adapted from selective_scan_fwd_kernel.cuh. The original implementation (in red) is replaced with an adjusted version (in green).

```
253  -   const float delta_a_exp = exp2f(delta_vals[i] * A_scaled)
254  +   const float delta_a_exp = 2.0f * exp2f(delta_vals[i] * A_scaled) - 1.0f
```

```
272  -   typename Ktraits::BlockScanT(smem_scan).InclusiveScan(
273  +   typename Ktraits::BlockScanT(smem_scan).ExclusiveScan(
274              thread_data, thread_data, SSMScanOp<weight_t>(), prefix_op
275          );
```

```
288  -   const float a = thread_data[i].y - (!kIsVariableB ? delta_vals[i] * float(u_vals[i]) :
289  -           delta_vals[i] * float(u_vals[i]) * B_vals[i]);
290  +   float delta_a_exp = 2.0f * exp2f(delta_vals[i] * A_scaled) - 1.0f;
291  +   const float ddelta_a_exp = delta_a_exp + 1;
292  +   const float a = ddelta_a_exp * thread_data[i].y;
293  +   const float hi = delta_a_exp * thread_data[i].y + (!kIsVariableB ? delta_vals[i] *
294  +           float(u_vals[i]) : delta_vals[i] * float(u_vals[i]) * B_vals[i]);
```

```
291  if constexpr (!kIsVariableB || !kIsVariableC) {
292      if constexpr (!kIsVariableB) { // dBC\_val is dB\_val
293  -       dBC_val += dout_vals[i] * (!kIsVariableC ? thread_data[i].y :  thread_data[i].y * C_vals[i]);
294  +       dBC_val += dout_vals[i] * (!kIsVariableC ? hi :  hi * C_vals[i]);
295      } else { // dBC\_val is dC\_val
296  -       dBC_val += dout_vals[i] * thread_data[i].y;
297  +       dBC_val += dout_vals[i] * thread_data[i].y;
298      }
299  }
300  if constexpr (kIsVariableB) { dB_vals[i] = dx * delta_vals[i] * float(u_vals[i]); }
301  if constexpr (kIsVariableC) {
302  -   dC_vals[i] = dout_vals[i] * (!kIsVariableB ? thread_data[i].y * B_val :  thread_data[i].y);
303  +   dC_vals[i] = dout_vals[i] * (!kIsVariableB ? hi * B_val :  hi);
304  }
```

Figure 12: Necessary changes to selective_scan_bwd_kernel.cuh. The original implementation (in red) is replaced with an adjusted version (in green).

```
196      if self.use_beta:
197  -   beta = rearrange(self.b_proj(hidden_states), 'b l h -> b h l').sigmoid()
198  +   beta = 2 * rearrange(self.b_proj(hidden_states), 'b l h -> b h l').sigmoid()
199      else:
200          beta = q.new_ones(q.shape[0], q.shape[1], q.shape[2])
```

Figure 13: Simple modification to the beta calculation in DeltaNet (Source) allowing the extension of the eigenvalues to the range $[-1, 1]$ . The original implementation (in red) is replaced with an adjusted version (in green).

