# OpenReview forum: "Unlocking State-Tracking in Linear RNNs Through Negative Eigenvalues"
_ICLR.cc/2025/Conference — ICLR 2025 Oral_

### Official Review · Reviewer_LYF4 · 2024-10-30

**Soundness:** 4
**Presentation:** 4
**Contribution:** 4
**Rating:** 8
**Confidence:** 4

**Summary:**

This paper investigates the expressivity of linear recurrent neural networks (LRNNs), which is a term synonymous with recent state space models (SSMs) and linear attention variants such as GLA. The paper proves a number of theoretical results on the representation power of different LRNNs with various structures on the transition matrices. First it shows several "necessary" theoretical results, such that simple language recognition problems such as parity and modulus counting can only be solved by LRNNs with more expressive state transitions, in particular those that extend the eigenvalue range from $[0,1]$ to $[-1, 1]$ or complex numbers. A number of "sufficient" theoretical results are also proven for the expressivity of products of generalized Householder matrices (i.e. those found in the recent DeltaNet model), showing its ability to recognize any regular language.

**Strengths:**

- The theoretical results are very comprehensive and illuminating.
- The paper's writing is very clear.
- There are empirical results on both these synthetic tasks (tracking the theory) as well as more end-to-end language modeling evaluations.

**Weaknesses:**

- No major weaknesses. Some minor additional inclusions could be helpful (see below)
- The paper's impact may be somewhat limited by the focus on synthetic tasks - it's not clear whether these actually matter in practice for real world tasks, restricting its applicability to large scale models on end-to-end modeling. E.g. these flavor of negative expressivity results for Transformers clearly haven't limited their applicability. I don't think this is a weakness of this particular paper, but a property of this broader research area, which prevents me from increasing my score.

**Questions:**

1. Do the models with this larger eigenvalue range run into more stability issues, especially at scale (e.g. the largest LM experiments)?
2. For Table 3, it may be nice to include a simple Transformer baseline.
3. Differences between models are sometimes conflated with differences in the neural network architecture (sometimes called the "block"). Meanwhile your results are about the inner recurrent layers and does not account for the block architecture. Does controlling for these two orthogonal factors affect results at all?
4. For Figure 4, does extending the eigenvalue range of Mamba also help on coding/math tasks?

---

> ### Author Response · Authors · 2024-11-19
>
> Dear Reviewer LYF4,
>
> We sincerely thank you for your thorough review of our submission.
>
> **Weakness:**
>
> We generally agree with the reviewer that the performance on synthetic tasks isn’t immediately indicative of real-world performance. However, we think that experiments on synthetic tasks provide a more scientific way of studying language models, and in many cases can be a good proxy for the real-world capabilities of large language models, as shown in Allen-Zhu Z. (2024). See also the second paragraph of the general response.
>
> **Questions:**
> > Stability issues:
>
> We have not observed any training instabilities, please see Figure 7 in the appendix. We similarly expected them to occur but this was not the case even for the 1.3B parameter models which we added for the rebuttal. We expect instabilities for long context lengths when the eigenvalue range is extended beyond $[-1, 1]$ and possibly also at higher learning rates.
>
> > Transformer baseline for Table 3:
>
> As you suggested, we have added a Transformer baseline in Tables 3, Table 5 and Figure 5 in the appendix. As expected, Transformers are not able to predict the correct token for lengths larger than the maximum context length seen during training. Thank you for the suggestion, as it underlines the limited ability of transformers to length-extrapolate in tasks from the Chomsky hierarchy.
>
> > On the impact of the block structure:
>
> Indeed, in our results we keep the block structure of Mamba 1 and DeltaNet fixed and only modify the inner recurrence. Changes to the block structure of Mamba and DeltaNet would not enable them to learn e.g. parity, but they might improve their general next-token prediction performance. An investigation into the block structure was done by [Gated DeltaNet](https://openreview.net/pdf/b4364be17e738609185d8d77f9c2ae800f22c28c.pdf) a concurrent ICLR submission.
>
> > Mamba for coding tasks:
>
> We show length extrapolation experiments for Mamba in Figure 8 in the appendix. Our modification generally degrades the performance of Mamba on CodeParrot and Math-Hard, as we discussed in the experimental section of the main paper. Interestingly, the perplexity of Mamba 370M on CodeParrot and Math-Hard is significantly higher than for DeltaNet 340M  (~32 vs ~23 for CodeParrot and ~32 vs ~21 for Math-Hard at 4096 tokens) while for TriviaQA and SlimPajama the perplexities for DeltaNet and Mamba are comparable. We hypothesize that these context retrieval and state-tracking heavy tasks are benefitting strongly from non-diagonal state-transition matrices of DeltaNet.
>
> Thank you again for your valuable input.
>
> **References:**
>
> Allen-Zhu Z. ICML 2024 Tutorial: Physics of Language Models

---

> > ### Comment · Reviewer_LYF4 · 2024-11-29
> > **Response**
> >
> > Thanks you for the clarifications. I will keep my score.

---

### Official Review · Reviewer_i3oU · 2024-11-01

**Soundness:** 3
**Presentation:** 3
**Contribution:** 4
**Rating:** 8
**Confidence:** 4

**Summary:**

### Summary of contributions

This paper studies the expressive capabilities of Linear Recurrent Neural Networks (LRNNs) on tasks requiring state-tracking. In particular, they look at the impact of the state-transition matrix structure which is usually a diagonal or a Generalized Householder (GH) matrix with positive values. They provide four theoretical findings promoting the use of non diagonal state-transition matrices with negative eigenvalues.

1) Building on the result from Sarrof et al. 2024 which states that LRNNs constrained to diagonal positive state-transition matrices cannot solve parity, they show the same is true for non-diagonal LRNNs.
2) Furthermore, to count modulo $m$ ($m\>2$), a LRNN (with finite precision and number of layers) requires at least one eigenvalue of the state-transition matrix to have modulus greater than or equal to one and with non-zero imaginary part.
3) One layer LRNNs\* can implement finite state-automata whose transition monoid are groups.
4) Multiple layer LRNNs\* can implement any FSA and thus can recognize regular languages.

\*(with transition matrix structured as product of sufficiently many GH matrices with eigenvalues ranging from \-1 to 1\)

These findings are supported by experiments comparing Mamba and DeltaNet, with and without negative eigenvalues. The results show that the architecture modifications suggested do not interfere with training and stability as well as confirming the applicability of these theoretical insights.

**Strengths:**

### Strong and weak points of the paper (list):

- The paper is well written and structured which enhanced clarity of the results and overall fluidity. In particular, the few lines introducing each section give a great summary of the key take-aways.
- The study of LRNNs expressivity, more precisely their capacity to achieve state-tracking, is well motivated through its connection to more complex and higher level problems such as reasoning and handling nested structures. The authors do contextualize their results by comparing and contrasting them with similar studies in the literature.
- The mathematical arguments presented in the paper are sound and rigorous which suggests the proofs (in appendix) are reliable. However, in some instances, more clarity should be provided (see questions below).
- Experiments supports not only the main claims of the theorems (e.g. Parity can be solved with negative eigenvalues, but not without), but also the more detailed insights (e.g. different levels of complexity explored in 5.2 State-tracking) and higher level conjectures (e.g. improved performance on language modeling tasks relying on state-tracking like coding). These experimental results make a compelling practical case for the modifications suggested by the theory in particular since the non-diagonal case is conclusive even with only one GH state-transition matrix (DeltaNet) while some theoretical results relied on a *product of $k$* GH matrices.

**Weaknesses:**

(see above)

**Questions:**

### Recommendation:

Accept

This work brings important clarification on the capabilities of LRNNs when given greater flexibility in their parameterization than the current widely used one. In particular, the result demonstrating the ability of LRNNs (with finite precision, depth and width) to recognize any regular languages independently of the sequence length is a strong one. Moreover, the experiments confirm that these findings do translate in practice without impacting training or stability.

### Questions (to clarify and increase confidence in the assessment):

- Some issues with the references:
  - 113:  Why referring to Liu et al 2023 for a tutorial on language theory and circuit complexity?  Did you mean to reference *Saturated Transformers are Constant-Depth Threshold Circuits* from Merrill et al 2022 (for circuit complexity)?
  - 116: Why citing Deletang et al. 2023 for Turing completeness of Transformers with arbitrary precision? Perez et al. 2021 seems to be the appropriate reference for arbitrary precision. *Theoretical Limitations of Self-Attention in Neural Sequence Models* by Hahn 2020 might be relevant for infinite depth.
- Could you explain how you obtained the equations for $\\mathbf{A(x\_t)}$ and $\\mathbf{B(x\_t)}$ for Mamba in Table 1? I am not able to come to this result from Algorithm 2 and the discretization (equation 4\) of Gu and Dao 2024\. Despite the mention that the matrices are applied to each row of the matrix $\\mathbf{H\_t}$, there needs to be more clarifications since the dimensions of the equations in Table 1 are in contradiction with the ones in equation (1) ($\\mathbf{A(x\_t)}$ is dxd vs nxn). This does not affect the theoretical results of the paper, but the experiments appear to rely on this definition given the chosen expression of $\\mathbf{s(x\_t)}$.
As suggested by the ICLR AI reviewer assistant, can you

>    1) Provide a step-by-step derivation of $\mathbf{A(x_t)}$ and $\mathbf{B(x_t)}$ for Mamba in Table 1.
>    2) Clarify how these equations relate to Algorithm 2 and equation 4 from Gu and Dao 2024.
>    3) Address the dimension mismatch between Table 1 and equation (1) for $\mathbf{A(x_t)}$.
>    4) Explain how this affects (or doesn't affect) the experimental results, particularly in relation to the expression of $\mathbf{s(x_t)}$.

- How does the comment on shortcut solutions (lines 220 to 222\) relate to the argument on finite precision? (The explanation as to why this example of recurrence can not be implemented in a finite precision LRNN is clear without this sentence.)
- 229: the statement *“so are unable to learn parity.”* in this context is too strong, the example simply proves that current diagonal LRNNs cannot implement *this* specific FSA for parity. To conclude they cannot solve parity would require a reference to Sarrof et al.
- 313: Why does the inclusion over the domain of eigenvalues imply inclusion over the sets of matrices (despite $k\neq k’$)? Is it implied that  $k=k’$?
- 314: Why mention that *“Repeated products of diagonal matrices… remain diagonal…”* given that GN matrices are not in general diagonal?

### Feedback to help improve the paper (not influencing assessment)

- 108 : Given the sentence *“Several studies have explored the expressive power of transformers”* a more comprehensive list would be appropriate. Some additional related references:
  - Separations in the Representational Capabilities of Transformers and Recurrent Architectures by Bhattamishra et al. 2024,
  - What Formal Languages Can Transformers Express? A Survey by Strobl et al. 2024
  - Simulating Weighted Automata over Sequences and Trees with Transformers by Rizvi-Martel et al. 2024
- Also, any reason why limiting to transformers and not mentioning the line of work on the expressive power of RNNs?
- 248 : Adding parenthesis in the counting modulo $m$ equation would be preferable to be consistent with line 212 and to prevent any ambiguity on operation priority.
- Figure 3 : Interesting that DeltaNet with 1 layer outperforms DeltaNet with 5 layers on S\_5 only swaps. Any intuition other than simple over parameterization?

---

> ### Author Response · Authors · 2024-11-19
>
> Dear Reviewer i3oU,
>
> We greatly thank the reviewer for the in-depth review and useful comments and corrections. In the following, we respond to each question in order.
>
>
> > Some issues with the references…
>
>
> Liu et al 2023 contains a short tutorial on groups and semigroups related to automata. We wanted to refer to it although it does not touch on circuit complexity theory. We changed the citation in the updated version to reflect this. The correct reference on arbitrary precision was indeed Perez et al. 2021, while we now replaced “infinite depth” with “arbitrary depth and weight sharing, citing the work on looped transformers.
>
>
> > Could you explain how you obtained the equations for  and  for Mamba in Table 1?...
>
>
> Thanks for pointing this out, we revised the equation in the updated paper, it was previously incorrect. To write the equation, we took inspiration from Table 4 in Yang et al. 2024 (where the state matrix is the transposed of ours). However, writing the appropriate state transition matrix becomes complicated due to the presence of the learnable parameter that we denoted with $W_1$, and would require concatenating the rows of the state (Yang et al. 2024 use the Hadamard product instead, which removes the issue). This is why we wrote the state-transition matrices for the recursion of each row (each being of dimension $d$ and hence the state-transition matrices will be $d \times d$). We note that In the original paper it is hard to find the true recursion for the Mamba model, also because the authors make some simplifications in the code (as mentioned in https://github.com/state-spaces/mamba/issues/19). A minimal version matching the original and which is fairly easy to read is provided in the mamba-minimal repo on github, which we also used to double check the correctness of our derivations. We verified that Table 4 in Yang et al. 2024 matches the mamba-minimal  implementation. The code we provide for the Modified Mamba Model is not affected by this issue.
>
>
> >How does the comment on shortcut solutions (lines 220 to 222) relate to the argument on finite precision?
>
>
> We agree that it is clear from the text that RNNs with finite precision cannot implement such solutions. The comment is meant to provide a similar conclusion for transformers, which might not be immediate for the reader. In addition, also the comment explains the term *shortcut solutions*, used by Liu et. al 2023 which is a well-known work in this research area.
>
>
> >Line 229: the statement "so are unable to learn parity."
>
>
> We added the citation to Sarrof et al. (2024) to provide more context for this statement.
>
>
> >313: Why does the inclusion over the domain of eigenvalues imply inclusion over the sets of matrices…
>
>
> We corrected the statement and changed it to: "Moreover, $\mathcal{M}\_{k}^n(\Omega)\subseteq\mathcal{M}_{k'}^n(\Omega')$ if $1 \in \Omega$, $k' \geq k$  and $\Omega \subseteq \Omega'$.” By requiring that $1 \in \Omega$ each element in the product of GH matrices can be the identity, this is why the statement is true for $k’ \geq k$ and not only when $k’ =  k$.
>
>
> >314: Why mention that “Repeated products of diagonal matrices… remain diagonal…
>
>
> We use this statement to contrast with the product of GH matrices. The reader might question if also the product of other types of matrices can be more expressive and we wanted to exclude the diagonal case. In the updated version, we replaced diagonal with triangular, since the statement is also true for this larger class of matrices. We also moved this part to the end of the paragraph to improve clarity.
>
>
> >108 : Given the sentence “Several studies have explored the expressive power of transformers” a more comprehensive list would be appropriate.
>
>
> Due to lack of space, we decided to focus on a subset of the works in this area which are most relevant to the paper. We added a remark for this in the updated version and referenced the survey paper you proposed.
>
>
> >any reason why limiting to transformers and not mentioning the line of work on the expressive power of RNNs?
>
>
> In the updated version, we changed the first phrase to also include RNNs and the cite relevant works.
>
>
> >248 : Adding parenthesis in the counting modulo …
>
>
> We added the parenthesis.
>
>
> > Figure 3 : Interesting that DeltaNet with 1 layer outperforms DeltaNet with 5 layers on S_5 only swaps. Any intuition other than simple over parameterization?
>
>
> In the updated experiments we use gradient clipping to further stabilize the training and pick the best of three seeds, in this case both one layer and 5 layers perform the same and length generalize perfectly up to 500 tokens (for all seeds).
>
>
> Thank you again for your thorough review of our work, please let us know if you have other questions or comments.

---

> > ### Comment · Reviewer_i3oU · 2024-11-25
> >
> > Great thanks for your reply and clarifications.
> > There still are inconsistencies in the Mamba equation (Table 1). Providing more clarification, even if only in appendix would be helpful (especially to build more theoretical work based on your paper).
> > - In the column for $A(x_t)$, a Hadamard product* is missing between the vectors $\Delta_t$ and $exp(\mathbf{w}_{1,i})$.
> > *Or any (vector, vector) $\to$ vector operation.
> >
> > - In the column for $\mathbf{B(x)}_t$, what operation is represented by the $\odot$ symbol? I assumed it was a Hadamard product, but $\mathbf{\Delta}_t \in \mathbb{R}^d$  while $\mathbf{x}_t \in \mathbb{R}^l$.
> > - In the column for dec$(\dots)$, what is(are) the dimension(s) of dec$(\dots)$ and $\mathbf{H}_t$? Is $\mathbf{q}_t^{\top} \mathbf{H}_t^{\top}$ an outer product (resulting in a matrix of dimension $d \times n$ if $\mathbf{H}_t$ is considered here a vector of dimension $d$)? As in the table column for $\mathbf{B(x)}_t$, what operation is $\mathbf{w}_2 \odot \mathbf{x}_t$? The two vectors have different sizes ($d$ and $l$).
> >
> > - 313: Ok. I had previously understood that the property might be true in two (less restrictive) cases :
> > 1. $1 \in \Omega$ and $k \geq k’$
> > 2. $k=k’$ and $\Omega \subseteq \Omega’$
> >
> > If it is not the case, please keep the correction as you did. If the previous statement is true however, it would be interesting to have the more general result.

---

> > > ### Author Response · Authors · 2024-11-26
> > >
> > > Thank you for the additional meticulous check of our work. We apologize for the mistakes in Table 1 that you correctly identified. We uploaded a revised version of the paper with the following corrections and changes.
> > >
> > > **Table 1:**
> > > 1. We added the Hadamard product between $\mathbf{\Delta}\_t$ and  $\exp(\mathbf{w}_{1,i})$.
> > > 2. We clarified that for Mamba $l=d$ so that now the dimension of $\mathbf{x}_t$ matches that of $\mathbf{\Delta}_t$ and $\mathbf{w}_2$.
> > > 3. We corrected the third column by replacing $\mathbf{q}_t^\top \mathbf{H}_t^\top$ by $\mathbf{H}_t^\top \mathbf{q}_t$ and specified that for Mamba, the matrices for the recursion on the columns $\mathbf{H}_t$ concern only the first two columns, i.e. $\mathbf{A}(\mathbf{x}_t)$, $\mathbf{B}(\mathbf{x}_t)$. In the third column, $\mathbf{H}_t \in \mathbb{R}^{n \times d}$ is the full state matrix and the output is a vector, since $\mathbf{q}_t \in \mathbb{R}^n$ and hence $ \mathbf{H}_t^\top \mathbf{q}_t \in \mathbb{R}^d$.
> > >
> > > **Line 313:**
> > > Indeed you are correct that the statement could be generalised by having two cases. We now say “Moreover, $ \mathcal{M}\_{k}^n(\Omega) \subseteq  \mathcal{M}_{k'}^n(\Omega')$ if $\Omega \subseteq \Omega'$ and either $k'=k$ or $k' \geq k$, $1 \in \Omega$.”
> > >
> > > **Appendix:**
> > > 1. We specified that with $\odot$ we mean the Hadamard product.
> > > 2. As you suggested, we added further clarification on the matrix-form of the Mamba recurrence as a new section in appendix A.2
> > >
> > > Thank you again for your detailed feedback, please let us know if you have any further comments.

---

### Official Review · Reviewer_tfg3 · 2024-11-07

**Soundness:** 4
**Presentation:** 4
**Contribution:** 3
**Rating:** 8
**Confidence:** 3

**Summary:**

This paper explores the connection between the eigenvalue range of the state transition matrix of a linear RNN and its ability to perform certain fundamental tasks, including parity and count modulo 3. The results point to changing the allowed eigenvalue range from [0,1] to [-1, 1] both for diagonal state transition matrices and ones constructed via identity plus a rank one component like DeltaNet. The authors show how to practically extend the eigenvalue range of both Mamba and Delta without harming training stability. Finally empirical results are shown on a 340M parameter DeltaNet and 370M parameter Mamba model with and without this eigenvalue range extension.

**Strengths:**

The paper provides a number of expressivity results which motivate considering state transition matrices with extended eigenvalue range. The first is an extension of previous work that for non-diagonal state transition matrices of a finite-precision LRNN, it still holds that solving parity for arbitrary input lengths requires at least one eigenvalue that is not real and positive (Theorem 1). Second, for modular counting with modulus greater than 2, at least one eigenvalue with a nonzero imaginary part is required to perform the task for arbitrary input lengths (Theorem 2). The paper then turns to practically how to extend the eigenvalue range for diagonal and generalized Householder state transition matrices (identity plus rank 1 component). Finally Theorem 3 and 4 provide two more expressivity results culminating in the  fact that "LRNNs" with state transition matrices that are repeated products of GH matrices each with eigenvalues in the range [-1,1] can recognize any regular language.

The work also provide a number of empirical results comparing Mamba and DeltaNet with eigenvalues in the range [0,1] and in the range [-1,1] as proposed by the paper. This includes looking at length generalization on a number of these more fundamental tasks discussed in the expressivity results, and the eigenvalue range is typically shown to be helpful. A 340M parameter DeltaNet and 370M parameter Mamba are then trained on 32B tokens of FineWeb with and without the eigenvalue range extension. I will discuss these experiments more in the next section.

Overall the paper is well-written, the results easy to follow, and the limitations clearly discussed. My confidence score reflects that I am not as familiar with related work in this area.

**Weaknesses:**

The main weakness of the results is that the performance of the eigenvalue range extension is somewhat mixed for the larger models discussed in Section 5.3. In particular, for the LM-harness benchmarks in Table 4, the accuracy is mostly at or below the original model. This means that the work may not have immediate impact on the SSM model architectures used at scale.

It would be very helpful in Table 4 to have the better accuracy/perplexity bolded for each model group. Figure 3 does not have the y-axis labelled.

**Questions:**

Do you any hypotheses for why for Mamba the [-1,1] model had worse perplexity on the datasets (Fig. 8) while for DeltaNet the [-1,1] model typically has better perplexity (Fig. 4)? More generally, when using this [-1,1] extension, would you have any recommendation for whether to use DeltaNet or Mamba?

---

> ### Author Response · Authors · 2024-11-19
>
> Dear Reviewer tfg3,
>
> Thank you for your thoughtful review of our paper. We appreciate your recognition of our work as well-written and easy to follow. We have carefully considered your concerns and questions and have addressed them as follows:
>
> > Regarding the lm-harness results reported in Table 4:
>
> We would like to emphasize that it wasn’t our aim to necessarily show improved results in Table 4 for models using the extended eigenvalue range, but mainly to show that the extended eigenvalue range does not lead to systematic loss in performance and demonstrate that we obtain comparable performance to the results reported by Yang et al. 2024. We only expected to obtain improved results for tasks that are related to state-tracking in some way for example in coding or in the step-by-step solutions which are part of math-hard shown in terms of perplexity in Figure 3. However, we agree with the reviewer that assessing the impact of our work in language modeling performance requires further research (see also the second paragraph of the general response).
>
> > It would be very helpful in Table 4 to have the better accuracy/perplexity bolded for each model group.
>
> Thank you for the suggestion, we have now updated Table 4 with bolded entries to improve readability.
>
> **Regarding your questions:**
> > Do you have any hypotheses for why for Mamba the $[-1,1]$ model had worse perplexity on the datasets (Fig. 8) while for DeltaNet the $[-1,1]$ model typically has better perplexity (Fig. 4)?
>
> Throughout the Chomsky hierarchy and state-tracking experiments on permutation groups, we saw limited gains for moving from $[0, 1]$ to $[-1, 1]$ for Mamba models for tasks more complex than parity. This is consistent with our language modeling results on code and math dataset which are linked to state-tracking, where extending the range seems to degrade performance.
>
> > More generally, when using this $[-1,1]$ extension, would you have any recommendation for whether to use DeltaNet or Mamba?
>
> We would recommend using DeltaNet over Mamba with the $[-1, 1]$ eigenvalue extension. We consistently see greater improvements for DeltaNet as it is a more expressive model due to its non-diagonal state-transition matrix. Unlike Mamba, this allows DeltaNet to perform simultaneous token- and channel-mixing, while Mamba can only perform token-mixing (in the recurrence). Furthermore, our theoretical results in Theorem 2, 3, 4 (and also in Appendix D of the updated version) show that the expressivity of DeltaNet $[-1,1]$ is greater than that of Mamba $[-1,1]$. We would like to mention that our change is also directly applicable to [Gated DeltaNet](https://openreview.net/pdf/b4364be17e738609185d8d77f9c2ae800f22c28c.pdf), which has a hybrid Mamba2/DeltaNet recursion. GatedDeltaNet is substantially better over the Mamba, DeltaNet and even a transformer baseline.
>
> If any points remain unclear, we are open to providing further explanation.
> Thank you for your time and valuable feedback.
>
> **References:**
>
> Yang, Songlin, et al. "Parallelizing Linear Transformers with the Delta Rule over Sequence Length." arXiv preprint arXiv:2406.06484 (2024).

---

> > ### Comment · Reviewer_tfg3 · 2024-11-25
> > **Response to Authors**
> >
> > Thanks to the authors for their response and paper revisions (I think Table 4 is much easier to parse with the modifications). Having read all the reviews and responses, I do not have any outstanding concerns and have thus raised my score from 6 to 8.

---

> > > ### Author Response · Authors · 2024-11-29
> > >
> > > Thank you for your feedback and for raising your score—we’re glad the revision improved clarity.

---

### Author Response · Authors · 2024-11-19
**General response to all reviewers**

We sincerely thank all reviewers for their thoughtful comments and suggestions, which helped us improve our paper. We are pleased that the reviewers acknowledge our theoretical results as novel and illuminating (Rev. LYF4), our experiments to be making a compelling case for the eigenvalue range extension (Rev. i3oU) and the results to be easy to follow (Rev. tfg3).

Rev. tfg3 and LYF4, both raised concerns about the potential limited impact of our results on language modeling. In response to that, we wish to emphasize that our research demonstrates that state-tracking is an area where efficient LRNNs can theoretically outperform Transformers. This finding opens up promising possibilities for developing LLMs and foundation models that learn solutions that are fundamentally different from the ones learned by Transformer models. These new models could potentially excel at reasoning tasks, rather than simply being more efficient but less capable versions of transformers. It is important to note that our proposed eigenvalue range extension makes current LRNNs strictly more expressive. Given the significant impact that LRNNs have already shown over the past years in architectures such as Samba (Sam et al. (2024)), Zamba (Glorioso et al. (2024)), and Codestral, we foresee great potential for further advancement in this field. We consider this a compelling research direction, even if future results turn out negative, as such findings would further validate the Transformer architecture's effectiveness.

We uploaded an updated version of the paper. In the following, we describe the changes from the previous version. Main changes are also highlighted in blue in the new version.

**C.1 Theory**:
After submission, we discovered an error in Theorem 2 and made a correction. While the theorem held true for a single layer, our analysis of multiple layers contained a flawed assumption: we incorrectly required that products of matrices with real eigenvalues would also have real eigenvalues. This is not true in general, as demonstrated by products of reflection (or Householder) matrices (see Proposition 1). In the updated statement of Theorem 2, *products* of state-transition matrices need to have complex eigenvalues (eigenvalues with nonzero imaginary part) with modulus greater than one in order to solve counting modulo m, and the number of elements in the product depends on the layer index. Therefore, for more than one layer, this negative result is not anymore true for our DeltaNet $[-1,1]$ model, which can have reflections as state transition matrices. Indeed, in the new Appendix D we prove an interesting result: a two layer LRNN using only reflections as state-transition matrices can implement addition modulo m. This shows that our modification is theoretically even more powerful than we previously thought, and all our major claims remain valid. We also would like to point out that we did not manage to provide a negative theoretical result on the expressivity of DeltaNet $[-1,1]$ with more than one layer, and we believe this presents an interesting future research direction as highlighted in the updated conclusions.

**C.2 Additional and Updated Experimental Results**:

*Language Modelling*:
After the submission we noticed that the results by Yang et al. (2024) to which we compare used the Mistral tokenizer while we used the Llama tokenizer. We therefore retrained DeltaNet (340M) and Mamba (370M) using the Mistral tokenizer and in the process also switched to the same training pipeline and data preprocessing used by Yang et al. (2024). Please see the updated appendix E.3.1 for training details. We also decided to train on more tokens (100B tokens of FineWeb) to test the model capacity so that the results are less limited by dataset size. Our updated results show improved performance for the models in Table 4 and the advantage of the extended eigenvalue range in Figure 3.

We provide additional results for DeltaNet with 1.3B parameters trained on 100B tokens of FineWeb in Table 4 showing comparable results to the DeltaNet model trained by Yang et al. (2024), again demonstrating that the eigenvalue range extension does not come at a cost of general performance. We provide new length extrapolation plots for the 1.3B parameter DeltatNet in Figure 3 showing that the advantage of the increased eigenvalue range on codeparrot and math persists, although with a diminished performance difference.

---

> ### Author Response · Authors · 2024-11-19
> **Continued response**
>
> *Chomsky Hierarchy*:
> Since we were unable to reproduce the results reported by Beck et al. (2024) on the Chomsky hierarchy for the initial submission, we performed additional experiments and found that gradient clipping and a lower learning rate were essential to improve training stability and final accuracy. Also, increasing the number of layers from 2 to 3 significantly improved the results for Modular Arithmetic without brackets, but only for DeltaNet $[-1,1]$. Through these changes, we found that DeltaNet $[-1, 1]$ and sLSTM reach respectively 97% and 78% sequence accuracy on modular arithmetic without brackets. We also saw significant improvements in performance for modular arithmetic with brackets. We present the updated results in Table 3 and Figure 4 in the appendix.
>
> *State-tracking*.
> We improved the experiments on the S5 variants (permutation composition) by training all models in all four setups, adding DeltaNet $[0,1]$ (1L), using gradient clipping to improve training stability, training each model for 100 epochs in all settings, and running 3 seeds instead of just one. The new Figure 2 is consistent with the old one and now generally shows an even larger gap between the models with and without our eigenvalue extension.
>
> **C.2 Other Changes**:
> Following the suggestions of the reviewers, we improved the related work section adding and correcting some references. We also made several minor revisions to improve clarity.
>
>
> **References:**
>
> Ren, L., Liu, Y., Lu, Y., Shen, Y., Liang, C., & Chen, W. (2024). Samba: Simple Hybrid State Space Models for Efficient Unlimited Context Language Modeling. arXiv preprint arXiv:2406.07522.
>
> Glorioso, P., Anthony, Q., Tokpanov, Y., Whittington, J., Pilault, J., Ibrahim, A., & Millidge, B. (2024). Zamba: A Compact 7B SSM Hybrid Model. arXiv preprint arXiv:2405.16712.
>
> Yang, Songlin, et al. "Parallelizing Linear Transformers with the Delta Rule over Sequence Length." arXiv preprint arXiv:2406.06484 (2024).
>
> Beck, M., Pöppel, K., Spanring, M., Auer, A., Prudnikova, O., Kopp, M., ... & Hochreiter, S. (2024). xLSTM: Extended Long Short-Term Memory. arXiv preprint arXiv:2405.04517.

---

> > ### Author Response · Authors · 2024-11-29
> >
> > We thank all reviewers for their valuable feedback, which has improved our work. If you have any further questions, we would be happy to address them during the remainder of the rebuttal period.

---

### Meta-Review · Area_Chair_Kkfc · 2024-12-20

**Metareview:**

The paper explores the approximation capabilities of linear RNNs including Mamba, mLSTM, and gated linear attention models. They find that incorporating negative values to state-transition matrices can significantly improve the expressivity of these models in solving parity and state tracking tasks. Besides enhancing expressivity, this theory-driven design also improves the empirical language modeling performance. Overall, all reviewers and I believe that this work provides new theory-driven design principles for linear RNNs and will make a strong addition to the conference program.

**Additional Comments On Reviewer Discussion:**

Several points were raised to improve the quality and results in the paper.

---

### Decision · Program_Chairs · 2025-01-22

Accept (Oral)